# Decadal fates and impacts of nitrogen additions on temperate forest carbon storage: A data-model comparison

Susan J. Cheng[1], Peter G. Hess[2], William R. Wieder[3,4], R. Quinn Thomas[5], Knute J. Nadelhoffer[6], Julius Vira[2], Danica L. Lombardozzi[3], Per Gundersen[7], Ivan J. Fernandez[8], Patrick Schleppi[9], Marie-Cécile Gruselle[10], Filip Moldan[11], Christine L. Goodale[1]

[1]Department of Ecology and Evolutionary Biology, Cornell University, Ithaca, NY, USA
[2]Department of Biological and Environmental Engineering, Cornell University, Ithaca, NY, USA
[3]National Center for Atmospheric Research, Boulder, CO, USA
[4]Institute of Arctic and Alpine Research, University of Colorado Boulder, Boulder, Colorado, USA
[5]Department of Forest Resources and Environmental Conservation, Virginia Tech, Blacksburg, VA, USA
[6]Department of Ecology and Evolutionary Biology, University of Michigan, Ann Arbor, Michigan, USA
[7]Department of Geosciences and Natural Resource Management, University of Copenhagen, Denmark
[8]Climate Change Institute and School of Forest Resources, University of Maine, Orono, ME, USA
[9]Swiss Federal Institute for Forest, Snow and Landscape Research, Birmensdorf, Switzerland
[10]Institute for Geography, University of Jena, Jena, Germany
[11]IVL Swedish Environmental Research Institute, Box 53021, SE-40014, Gothenburg, Sweden

*Correspondence to*: Susan J. Cheng (sjc265@cornell.edu)

**Abstract.** To accurately capture the impacts of nitrogen (N) on the land carbon (C) sink in Earth system models, model responses to both N limitation and ecosystem N additions (e.g., from atmospheric N deposition and fertilizer) need to be evaluated. The response of the land C sink to N additions depends on the fate of these additions—that is, how much of the added N is lost from the ecosystem through N loss pathways, or recovered and used to increase C storage in plants and soils. Here, we evaluate the C-N dynamics of the latest version of a global land model, the Community Land Model version 5 (CLM5), and how they vary when ecosystems have large N inputs and losses (i.e., an open N cycle) or small N inputs and losses (i.e., a closed N cycle). This comparison allows us to identify potential improvements to CLM5 that would apply to simulated N cycles along the open-to-closed spectrum. We also compare the short- (< 3 years) and longer-term (5-17 years) N fates in CLM5 against observations from 13 long-term $^{15}$N tracer addition experiments at eight temperate forest sites. Simulations using both open and closed N cycles overestimated plant N recovery following N additions. In particular, the model configuration with a closed N cycle simulated that plants acquired more than twice the amount of added N recovered in $^{15}$N tracer studies on short timescales (CLM5: 46 ± 12%; observations: 18 ± 12%; mean across sites ± 1 standard deviation), and almost twice as much on longer timescales (CLM5: 23 ± 6%; observations: 13 ± 5%). Soil N recoveries in simulations with closed N cycles were closer to observations on the short (CLM5: 40 ± 10%; observations: 54 ± 22%) term, but smaller than observations on the longer-term (CLM5: 59 ± 15%; observations: 69 ± 18%). Simulations with open N cycles estimated similar patterns in plant and soil N recovery, except that soil N recovery was also smaller than observations on the short-term. In both open and closed sets of simulations, soil N recoveries in CLM5 occurred from the cycling of N through plants rather than through direct immobilization in the soil, as is often indicated by tracer studies. Although CLM5 greatly overestimated plant N recovery, the simulated increase in C stocks to recovered N was not much larger than estimated by observations, largely because the model's assumed C:N ratio for wood was nearly half that suggested by measurements at the field sites. Overall, results suggest that simulating accurate ecosystem responses to changes in N additions requires increasing soil competition for N relative to plants, and examining model assumptions of C:N stoichiometry—which should also improve model estimates of other terrestrial C-N processes and interactions.

**1 Introduction**

Biogeochemical processes in plants and soils influence Earth's climate by controlling how much carbon dioxide ($CO_2$) can be removed from the atmosphere and placed into long-term storage in terrestrial ecosystems (Bonan, 2008). Currently, Earth system model ensembles that compare multiple models against each other persistently show a large uncertainty around estimates of $CO_2$ fluxes exchanged between the land surface and the atmosphere under future scenarios of increasing $CO_2$ and climate change (Friedlingstein et al., 2006; Anav et al., 2013; Friedlingstein et al., 2014). This uncertainty is mainly driven by differences in how models represent biological processes on land and their responses to increasing atmospheric $CO_2$ concentrations (Lovenduski and Bonan, 2017; Bonan and Doney, 2018). Nutrient limitation is one factor that is likely to constrain $CO_2$ fertilization of the biosphere (Zaehle and Dalmonech, 2011; Medlyn et al., 2015; Wieder et al., 2015b; Zaehle et al., 2015; Meyerholt and Zaehle, 2018), but not all global land models used in coupled climate-land simulations include explicit representations of the nitrogen (N) cycle. As more global land models add and examine the impacts of coupled carbon (C) and N cycles (Thornton et al., 2007; Sokolov et al., 2008; Zaehle and Friend, 2010; Smith et al., 2014; Goll et al., 2017), additional sources of uncertainty will be added into these models (Wieder et al., 2015a; Lovenduski and Bonan, 2017). Currently, some of the principal uncertainties in simulating terrestrial C-N interactions lie in how models represent plant acquisition of N from soil and the relative competitiveness among plants, decomposers, and denitrifying microbes for soil N (Thomas et al., 2013b; Medlyn et al., 2015; Thomas et al., 2015; Zhu et al., 2016b). Thus, evaluating model representations of N cycling is critical for improving our understanding of the magnitude of ecosystem C response to changes in N additions (dC/dN; Sutton et al. (2008)) and how dC/dN influences the size of the terrestrial C sink over the 21$^{st}$ century.

Human uses of fossil fuels, N-fixing plants, and fertilizers have more than doubled rates of N additions to terrestrial ecosystems compared to preindustrial conditions (Vitousek et al., 1997; Galloway et al., 2003). Increased emissions of reactive N from combustion processes and agricultural sources have led to increases in atmospheric N deposition (Galloway et al., 2003; Vet et al., 2014), which can have multiple effects on forests and other terrestrial ecosystems (e.g., Aber et al. 1998). These effects include shifts in rates of tree growth (Solberg et al., 2009; Thomas et al., 2010) and soil decomposition (Janssens et al., 2010; Frey et al., 2014), as well as increased soil emissions of nitrous oxide (Butterbach-Bahl et al., 2002). The fate of N deposition in plants, soils, or N loss pathways from forests is central to quantifying the effect of N deposition on terrestrial C storage (Emmett et al., 1998a; Nadelhoffer et al., 1999a; Currie et al., 2004; Lu et al., 2010; Templer et al., 2012; Lovett et al., 2013; Wang et al., 2018). Woody plant tissues have higher C:N ratios (e.g., 100-500) than foliage and roots (e.g., 20-40), which allow trees to build more organic C per unit N taken up by plants compared to other plant types. Similarly, woody tissues have C:N ratios that are one to two orders of magnitude higher than soil organic matter (e.g., 5-25) (Nadelhoffer et al., 1999b; Yang and Luo, 2010; Zechmeister-Boltenstern et al., 2015; Goodale, 2017), allowing trees to store much more additional C if they successfully compete for N deposition than if N is retained in soil; no additional C is stored in forests when N is lost from the system by denitrification or N leaching (Nadelhoffer et al., 1999b).

The fate of N deposition in terrestrial ecosystems has been quantified through field tracer experiments that apply a small amount of highly enriched N with its stable isotope ($^{15}$N) to the forest and subsequently measure the recoveries of that $^{15}$N tracer in plant and soil pools. Reviews of these $^{15}$N experiments, which are located predominately in North America and Europe (Tietema et al., 1998; Nadelhoffer et al., 1999b; Curtis et al., 2011; Templer et al., 2012), as well as in warm and humid sites in China (Gurmesa et al., 2016; Wang et al., 2018), indicate that the total amount and partitioning of recovered $^{15}$N varies across sites, but that litter and soil pools typically dominate as sinks for N additions during the first few years after tracer application. These litter and soil $^{15}$N sinks often occur directly through microbial or chemical processes within days or weeks after tracer application, without first passing through plants (Berntson and Aber, 2000; Perakis and Hedin, 2001; Providoli et al., 2006; Lewis and Kaye, 2012; Goodale et al., 2015). These $^{15}$N tracer studies are also useful for quantitatively evaluating coupled C-N cycle processes in land models. For example, Thomas et al. (2013b) used mean results from short-term tracer experiments (< 3.5 years) to test the responses of two coupled C-N models that treat plant and soil responses to N additions differently: O-CN (Zaehle and Friend, 2010) and the Community Land Model version 4 (CLM4) (Thornton et al., 2007). That analysis showed that CLM4 lost a large fraction of incoming N additions to N gases, while the N retained in the ecosystem was distributed relatively evenly between plants and soils. In contrast, OCN better estimated total ecosystem retention of N additions, and projected that soils dominated the short-term fate of added N.

Comparisons of model simulations with long-term field experiments ultimately provide more relevant constraints on decadal or centennial-scale forest C uptake and N cycling dynamics than comparisons using short-term field studies that often reflect transient dynamics (Perakis and Hedin, 2001; Jefts et al., 2004; Providoli et al., 2006; Templer et al., 2012). For example, field studies indicate that N initially retained in litter and soil could redistribute to plants and enable additional C uptake over the long-term; alternatively, retained N could accumulate in soil pools or be lost from the ecosystem entirely (Nadelhoffer et al., 2004; Krause et al., 2012; Wessel et al., 2013; Goodale, 2017). However, the long-term fates of N deposition in land models have not yet been evaluated against a synthesis of field measurements. This is in part because, to date, there have only been a handful of individual site-level field studies published that have examined the long-term fates of N additions (e.g., Nadelhoffer et al. (2004) and Krause et al. (2012)). This study addresses this gap by compiling a summary of $^{15}$N recovery data from long-term $^{15}$N tracer experiments. We then use these data to evaluate the capability of an updated version of the Community Land Model (i.e., CLM5; Lawrence et al. (in review))—the land component of the Community Earth System Model that will be part of the Coupled Model Intercomparison Project phase six (CMIP6)—in its ability to accurately simulate the impacts of ecosystem N additions on annual to decadal timescales. CLM5 includes new, more mechanistic representations of plant N processes, following earlier changes to soil C-N dynamics in CLM4.5 (see Section 2.2), which could affect the fate and impact of N in ecosystems. Changes in the fate of N in ecosystems could also be a result of how strongly an ecosystem retains N additions, which is a function of the rates of N added and lost from the ecosystem, as well as internal N cycling dynamics (Aber et al., 1998). To identify whether model-simulated N fates depend on the magnitude of N fluxes, we also compared N fates in CLM5 using two kinds of N cycles—one with high rates of N inputs and losses, and one with lower rates of N inputs and losses. This comparison also allowed us to identify potential improvements to the model that are independent of N fluxes. Through this

novel data-model comparison project, we provide a synthesis of long-term, ecosystem $^{15}$N addition experiments—and identify how differences in temporal dynamics of N cycling between field measurements and CLM5 lead to divergences in measured and modeled N fate and ecosystem C responses to N additions.

**2 Methods**

To assess CLM5's ability to accurately simulate C-N dynamics on both short and longer timescales, we first compiled existing and newly available field data from eight sites that applied a $^{15}$N tracer at least a decade ago. We ran model simulations for each site to examine how N fates and C sink responses to N additions might differ in simulated ecosystems with high N inputs and losses (characteristic of an open N cycle) and simulated ecosystems with low N inputs and losses (characteristics of a closed N cycle). Details on which fluxes the model includes as N inputs and losses are described in Section 2.2. Comparing the open and closed representations of the N cycle, described in more detail in Section 2.3.1, allows us to examine how sensitive CLM5 is to the "openness" of an ecosystem's N cycle. We also evaluate how the model's C sinks respond to N deposition on both the short-term and longer-term. We define short-term recovery as time points within 3 years after the tracer was applied because the majority of rapid changes in modeled N recovery occur during this time (see Results) and many $^{15}$N experiments report results within 1-3 years after tracer application (Templer et al., 2012). Longer-term recovery includes time points after 3 (i.e., 5 –17) years.

**2.1 $^{15}$N tracer field experimental sites**

At each of the eight field sites used to evaluate CLM5 (Table 1), a $^{15}$N tracer was added at least 10 years ago, often under both ambient and fertilized conditions. These sites span a range of environmental conditions in North America and Europe, and include two plant functional types (PFTs) in CLM5: broadleaf deciduous temperate (BDT) and needleleaf evergreen temperate (NET) trees. Present-day ambient N deposition at these sites ranges from approximately 0.8 to 2.0 g N m$^{-2}$ y$^{-1}$. Across sites, a $^{15}$N tracer was added—as either ammonium, nitrate, or in some cases, in both forms—to five $^{15}$N experiments under ambient conditions and to eight experiments under fertilized conditions, with additions ranging from 2.5 to 7.5 g N m$^{-2}$ y$^{-1}$. Available field measurements of $^{15}$N recovery from these sites are in SI Table 1 and available at https://doi.org/10.5281/zenodo.2772160.

**2.2 Model description**

We evaluated CLM5 (development version 16_r253) in both its ability to estimate the site-level fate of N deposition against the eight experimental sites listed in Table 1. CLM5 is the terrestrial component of the Community Earth System Model (CESM 2.0) and has undergone several changes to its C and N biogeochemistry since CLM4. Briefly, in CLM4.5, the model's original soil biogeochemistry was replaced with a vertically-resolved CENTURY-based approach

and is described in detail by Koven et al. (2013) and Oleson et al. (2013). In CLM5, three important changes were made to plant C and N dynamics. First, the Leaf Utilization of Nitrogen for Assimilation (LUNA) module allows plants to adjust their photosynthetic capacity (i.e., the maximum rate of carboxylation; $V_{c,max}$) based on environmental conditions (Ali et al., 2016). Specifically, $V_{c,max}$ is influenced by the amount of leaf N allocated for carboxylation, as well as day length and season. Second, plants can alter and optimize their stoichiometry (FlexCN module), which removed the down-regulation of gross primary productivity (GPP) that was used in CLM4 and CLM4.5 (Ghimire et al., 2016). The amount of N that is allocated to individual sub-plant pools is determined based on a fixed set of allometric ratios and the amount of N the plant has for new growth. Additional details on how stoichiometry is optimized can be found in the CLM5 documentation referenced below. Third, in the Fixation and Uptake of Nitrogen (FUN) module, plants pay C costs (which are respired) for acquiring N from symbiotic N fixation, uptake of soil N, and retranslocation (Shi et al., 2015). Additional information about these modifications, as well as other changes to model processes and parameterizations can be found in the model documentation (Lawrence et al., 2018).

As in prior versions of the model, C and N cycles in CLM5 are coupled at 30-minute time steps through plant and soil competition for soil N and internal recycling of plant and soil material through litterfall (Thornton et al., 2007; Koven et al., 2013; Oleson et al., 2013; Thomas et al., 2013a). New N inputs enter an ecosystem through N deposition, and free-living and symbiotic N fixation. When N deposition is added to the inorganic soil N pool, it is distributed vertically through the soil column according to an exponential profile; approximately 40% of N deposition is added to the top 2 cm and approximately 95% is added to the top 20 cm. When the amount of soil N is smaller than the total N demand, soil N is divided between plants and an implicit representation of microbial immobilization into soil and litter based on each N sink's proportionate demand to the total N demand. Free-living biological N fixation is calculated as a function of annual evapotranspiration and added to the soil mineral N pool. Symbiotic N fixation is passed directly to the plant and depends on plant N demand, the cost of N fixation for the plant, and soil temperature (Lawrence et al., 2018; Lawrence et al., in review); details on the model's representation of N fixation is available at https://escomp.github.io/ctsm-docs/doc/build/html/tech_note/FUN/CLM50_Tech_Note_FUN.html. Subsequent losses of N occur through production of N gases during nitrification and denitrification. Denitrification occurs in the anoxic portion of the soil and is constrained by decomposition and the availability of nitrate. After gaseous losses, N is lost through water—specifically through surface runoff of dissolved inorganic N over land to stream flow and sub-surface leaching through the soil column; the model does not simulate losses of organic N. Rates of decomposition are limited by soil moisture, soil temperature, oxygen availability, and N availability. As litter decomposes into soil organic matter, a portion of C is respired and N is transferred from litter pools through to soil pools. In all our simulations (described below), we turned off transient losses of biomass N from fire and harvest because these disturbances infrequently occur at the sites we simulated.

### 2.3  Model simulations

For each site and N cycle configuration (see 2.3.1 below), initial ecosystem C and N stocks for 1850 were generated using a spin-up approach where the model was run using 1850 concentrations of $CO_2$ (285 ppm) and the model's standard climate forcing dataset from the Global Soil Wetness Project Phase 3 (GSWP3; http://search.diasjp.net/en/dataset/GSWP3_EXP1_Forcing) developed by Kim (2017). Specifically, we ran the model for 500 years in accelerated decomposition mode by cycling through the 1901-1920 climate forcing dataset, and then for a minimum of 1500 years in regular mode until soil and plant C and N stocks achieved steady state. Subsequently, we ran a historical simulation from 1850 to 2010 (or until 2015 for Arnot Forest, where [15]N recovery was measured after 2010) using transient GSWP3 climate, N deposition, and atmospheric $CO_2$. We ran CLM5 in single-point mode for each site, and modified each site's grid cell area to contain a single PFT and land surface unit (e.g., no lakes). Finally, following Thomas et al. (2013a), we implemented a harvest in the year that established the observed, present-day stand age for each site. For Arnot Forest, simulations cycled through the 2006-2010 forcing data for the model years 2011-2015.

### 2.3.1 Modeling open and closed N cycles in CLM5

Similar to prior versions of the model, CLM5 uses input data for N deposition and models fluxes of other N inputs and losses that are unrealistically high for temperate forests (Thomas et al., 2013a; Thomas et al., 2013b). Prior versions of CLM also greatly underestimate measured rates of N losses to leaching and runoff in these ecosystems (e.g., MacDonald et al. (2002), Aber et al. (2003), Nevison et al. (2016), Thomas et al. (2013a)). To address the differences between fluxes used in CLM5 and field-based expectations (see below; Fig. 1), we simulated how each of the field sites in Table 1 would respond to N deposition under a) CLM's default N cycle with high N inputs and losses (i.e., open N cycle), and b) under an alternative "closed" version of the model where N inputs were reduced to better match observations and N losses to denitrification were correspondingly halted to allow the model to build realistic soil N pools and N leaching losses (see Section 3.1). In CLM5's configuration with an open N cycle, N deposition rates come from atmospheric modeling simulations described in Lamarque et al. (2010). In the year 1850 (Fig. 1a), the N deposition rate in CLM5 averaged across our sites was $0.4 \pm 0.3$ g N $m^{-2}$ $y^{-1}$ (mean $\pm$ 1 standard deviation; SD across modeled sites), exceeding the estimated rate of about 0.1 g N $m^{-2}$ $y^{-1}$ for pre-industrial times (Holland et al., 1999; Fakhraei et al., 2016). The temperate forests we simulated also rarely contain plants with N-fixing symbionts, and have rates of free-living N fixation closer to around 0.1 g N $m^{-2}$ $y^{-1}$ (Vitousek et al., 2013; Tedersoo et al., 2018). However, CLM5 has symbiotic N fixation rates in 1850 of $0.5 \pm 0.3$ g N $m^{-2}$ $y^{-1}$ and mean free-living N fixation of $0.3 \pm 0.03$ g N $m^{-2}$ $y^{-1}$. In addition, losses of N to denitrification were as high as previously found in CLM4 and CLM4.5 (Thomas et al., 2013a; Houlton et al., 2015), accounting for 99% of all N losses (Fig. 1b).

In the configuration of CLM5 with a closed N cycle, we changed the model's ecosystem N inputs to be more consistent with historical reconstructions and measurements that suggest 19th century NOx emissions were smaller (Hoesly et al., 2018) than those estimated by Lamarque et al. (2010). To achieve this, we first lowered pre-industrial N deposition to 0.1 g N m$^{-2}$ y$^{-1}$ for the year 1850 (Fakhraei et al., 2016) and to 0.2 g N m$^{-2}$ y$^{-1}$ in 1950 to account for the doubling in N deposition and NOx emissions from widespread use of fertilizers produced by the Haber-Bosch process (Galloway et al., 2003; Reay et al., 2008). Then we set N

deposition from 1975 to present to the site-reported N deposition rate (Engardt et al., 2017) in Table 1, and used linear interpolation to calculate N deposition between 1850-1950 and 1950-1975. After 1975, N deposition was held constant to roughly match regional trends (Driscoll et al., 2003; Galloway et al., 2013). These changes allowed us to better simulate the expected N deposition rates at our modeled sites so that the effect of N deposition on N fates between CLM5 and field experiments are more comparable. To adjust N fixation rates, we turned off symbiotic N fixation and set free-living N fixation at all sites to 0.1 g N m$^{-2}$ y$^{-1}$ to match field expectations (Cleveland et al., 1999; Tedersoo et al., 2018). With these dramatically lower rates for N inputs (Fig. 1a, left panel), simulated C and

N stocks were too small, which required us to turn off N losses from denitrification to achieve realistic baseline plant and soil C stocks (Table 3; Section 3.1). With denitrification turned off, hydrologic losses of N increased so that the model's present-day sub-surface leaching and runoff fluxes each increased to around 0.1 g N m$^{-2}$ y$^{-1}$ (Fig. 1b, right panel), which is within the range measured in temperate forest stream water (Aber et al., 2003; Gundersen et al., 2006; Groffman et al., 2018). A model that simulates small fluxes of both heterotrophic N fixation inputs and denitrification losses might best match observations of these processes in these temperate forests (Tjepkema, 1979; Roskoski, 1980; Hendrickson, 1990; Barkmann and Schwintzer, 1998; Bernal et al., 2012; Duncan et al., 2013;

Morse et al., 2015), but requires substantial model development to achieve (Thomas et al., 2013a; Houlton et al., 2015). Our alternative model included in this study is thus an oversimplification included to examine model C-N responses in ecosystems with a much more closed N cycle than released versions of CLM5, details of which can be found in Lawrence et al. (2018). Our results below highlight how sensitive CLM5 is to the openness of its N cycle, an emergent property that should be the focus of future model development.

**2.4 Model N recovery calculations**

To calculate N recoveries in ecosystems under ambient N deposition in CLM5, we followed the approach of Thomas et al. (2013b) of adding a small amount of additional N into the N deposition input stream, which enters the model's inorganic soil pool as $NH_4^+$. Across the field studies, $^{15}N$ tracers were applied differently, with variation in N addition rates, forms ($^{15}NH_4^+$ or $^{15}NO_3^-$), and timing (Table 1), although most tracer applications were distributed across the growing season. For our simulations under ambient N deposition, we implemented a consistent approach across all simulations by adding a 0.5 g N m$^{-2}$ y$^{-1}$ "tracer" in the model during the first year the tracer was applied in the field. In the first year, we applied the N "tracer" in the model equally across days during

April through September to capture the most active portion of the growing season. In CLM5, we ran sensitivity tests for two of our eight model sites (see Table 1 for a full list, including site names), an old growth forest (Alptal) and a younger forest (Harvard NET), which confirmed that the smallest amount of N we could

apply while maintaining realistic ecosystem N recovery responses at both sites was 0.5 g N m$^{-2}$ y$^{-1}$ (Fig. S1), and is consistent with Thomas et al. (2013b). A sensitivity test for Harvard NET also indicated that the mean N recovery across the last 20 years of the historical simulation was relatively insensitive to which months within the growing season the N "tracer" was applied (Fig. S2). For simulations with fertilization, we applied the site-reported fertilization rate (Table 1) in the model during all years the fertilizer was applied in the field—but only during April through September of each of those years, as we did for simulations

under ambient N deposition.

       Annual N recovery was then calculated for each N cycle configuration, site, and year by taking the difference in N stocks between a baseline simulation without a "tracer" or fertilization treatment and its corresponding simulation with "tracer" or fertilizer added, according to:

$$N\ recovery\ (t) = \frac{N\ stock(t)_{N\ addition} - N\ stock(t)_{baseline}}{\sum_0^t N\ inputs_{N\ addition} - \sum_0^t N\ inputs_{baseline}} \qquad (1)$$

where $N\ stock(t)_{N\ addition}$ is the N addition simulation's N stock at year $t$ since the application of the "tracer" or fertilizer, $N\ stock(t)_{baseline}$ is the N stock from the simulation without a "tracer" or fertilizer added, and $N\ inputs_{N\ addition}$ and $N\ inputs_{baseline}$ are the total amounts of N entering the ecosystem in the N addition and baseline simulations at each time, respectively. N inputs are the sum of N deposition, the added "tracer" or fertilizer, and biological and free-living N fixation. Total vegetation stocks in the model include N in all aboveground and belowground plant pools, including plant stem, foliage, fine roots, and coarse roots. Soil N

stocks include N in litter, organic matter, and soil inorganic N pools. We did not include coarse woody debris in the soil stock because coarse woody debris is rarely measured in $^{15}$N tracer experiments. Simulated N recovery in coarse woody debris is possible in CLM5, although this pool accounted for less than 3% of recovered N. Thus, we applied the field definition of total ecosystem recovery to the model results as well—defining total ecosystem recovery in the model as the sum of the plant and soil pools. We note that N recoveries of N deposition or fertilizer in ecosystems simulated with open N cycles (reported in Table S1) include a small effect of N fixation rates changing in response to added "tracer" or fertilizer. Calculated recoveries in simulated ecosystems with closed N cycles (shown

in the Results) are a response to the added "tracer" or fertilizer alone, because the control and N addition simulations have the same, fixed N-fixation rates.

**2.5 Calculating ecosystem C response to N additions**

       To examine the impact of model-estimated N fates on the land C sink, we calculated the modeled change in plant or soil C storage per unit change in N input (g C g$^{-1}$ N; i.e., dC/dN), which is frequently done to quantify the impacts of N additions on ecosystem C pools (De Schrijver et al., 2008; Sutton et al., 2008; Thomas et al., 2013a; Frey et al., 2014). Model dC/dN (g C g$^{-1}$ N) for each C pool of interest (e.g. total soil or total plant pool) was calculated for each year

according to:

$$\frac{dC}{dN}(t)_{model} = \frac{C\ stock(t)_{N\ addition} - C\ stock(t)_{baseline}}{\sum_0^t N\ inputs_{N\ addition} - \sum_0^t N\ inputs_{baseline}} \qquad (2)$$

where $C\ stock(t)_{N\ addition}$ is the mean C plant or soil stock at time $t$ in the N addition simulation since the application of the "tracer" or fertilizer, and $C\ stock(t)_{baseline}$ is the mean C plant or soil stock at time $t$ in the baseline simulation since the application of the tracer in the N addition simulation. Alternatively, we can estimate both the field and model dC/dN using the method from Nadelhoffer et al. (1999b):

$$\frac{dC}{dN}(t) = \sum_{i=1}^n N\ recovery_i \times C{:}N_{pool(i)} \qquad (3)$$

where $N\ recovery$ is the N recovery of the $i$th component of a measured plant or soil stock (e.g., foliage, wood, litter), $C{:}N_{pool(i)}$ is the C:N ratio of the $i$th component that makes up a particular stock, and the summation is over $i$ pools that make up the total plant or soil stock. We based our C:N ratios for stocks on site-reported values from the literature (see Table 2). Ecosystem dC/dN is the sum of plant and soil dC/dN.

**2.6 Statistical analyses**

To determine whether model estimates of ecosystem traits, N recovery, and dC/dN differ from observations, we used one-sample t-tests to identify whether the differences between the model and field values were significantly different from 0. For all statistical analyses, we used the Shapiro-Wilk test to check for normality. When statistical assumptions were not met, we tested for differences using a one-sample Wilcoxon rank sum test.

**3    Results**

Below, we first report the effects of N cycle openness (i.e., high or low N inputs and losses) on modeled ecosystem C and N stocks (Section 3.1). Next, we compare model output to available observations for a) the change in N recovery over time at individual sites (Section 3.2), b) the mean change in N recovery in plant and soil pools on short- (< 3 years) and longer (> 3 years; 5 to 17 years) timescales for all tracer experiments in this study (Section 3.3.1), c) the estimated impacts of these N fates on C stocks (i.e., dC/dN; Section 3.3.2), and finally, d) the patterns in N recovery that emerge by PFT and fertilizer treatment (Section 3.4).

**3.1 N cycle openness: Comparison of CLM5 with open and closed N cycles**

Ecosystem traits from the end of the historical simulations are reported in Table 3, along with available site measurements. Simulations using a relatively open N cycle had different rates of N inputs and losses in 1850 and throughout the historical simulation than simulations using a relatively closed N cycle (Fig. 1). Observed soil C stocks were typically higher than modeled under both open and closed N cycles in CLM5 ($p < 0.01$, Table 3). However, simulations with both an open and closed N cycle produced present-day aboveground net primary productivity rates (ANPP), leaf area index, plant C stocks, and plant and soil N stocks that were statistically similar to observations ($p > 0.05$, Table 3).

The two model configurations differed by an order of magnitude in their estimated ecosystem N turnover time (i.e., pool size of the ecosystem divided by total N loss fluxes from denitrification, leaching, emissions of $N_2O$ from nitrification, and runoff). Simulations using a closed N cycle had a higher mean ecosystem N turnover time of $6500 \pm 5300$ years (mean $\pm$ 1 SD across simulated sites) compared to $880 \pm 370$ years in simulations using an open N cycle (Table S2). The wide range in ecosystem N turnover time results from site-specific differences, including variations in factors governing organic N storage in soils (e.g., texture, past disturbance). By comparing the ratio between the mean turnover time of N in the ecosystem to the mean turnover time of N in plants—where plant N turnover time was calculated as the size of the plant N pool divided by the loss of N from the plant pool from litterfall—we quantified how frequently N could cycle through the plant pool. This metric describes the potential for a unit of N to be used by plants to produce C before it is lost from the system. In simulations with a closed N cycle, ecosystem N cycled through plants an average of $210 \pm 190$ times before it was lost, while it only cycled through plants $26 \pm 15$ times in simulations with an open N cycle. Consequently, the same unit of N in a closed N cycle has a longer retention time in plants than when the N cycle is open— which could lead the model configuration with a closed N cycle to produce more plant and soil C per unit of N. Given high biases in the historical N deposition, biological N fixation, denitrification, and leaching/runoff rates in the model with an open N cycle (see Section 2.3.1), we focus the remainder of the study on results from the model configuration with a closed N cycle. For completeness, results of N recovery for CLM5 using an open N cycle, which is the default configuration of N cycling in CLM5, are in Table S1 and at https://doi.org/10.5281/zenodo.2772160.

**3.2. Site-level decadal changes in N recovery**

The model-measurement comparisons of N recovery for each site, experimental N treatment, and N cycle configuration are given in Table S1 and shown in Fig. 2 and Figs. S3-S6 (data in Table S1 are also available at https://doi.org/10.5281/zenodo.2772160). In this section, we highlight the recovery at Harvard Forest BDT (deciduous) and Harvard Forest NET (evergreen) under ambient N deposition because Harvard Forest provides the longest record for a model-measurement inter-comparison with [15]N tracer results. Under a closed N cycle, both Harvard Forest PFTs simulated that plants were the dominant, immediate fate of this added N, accounting for 63% of added N in the BDT stand and 49% in the NET stand in the first year after N addition. In contrast, field

measurements demonstrate that plants acquired < 10% of added tracer, and that soils were the dominant sink for $^{15}$N (Fig. 2; Nadelhoffer et al. (2004)). After the first year, the model estimated that the N initially taken up by plants moved to soils within 3 years in the deciduous stand and within 5 years in the evergreen stand, after which the recovered N subsequently stayed in the soil pool—accounting for approximately 70% of the added N at the end of two decades. This pattern was typical across the sites we simulated and across ecosystems under open and closed N cycles. However, total ecosystem recovery was generally lower in ecosystems with an open N cycle relative to ecosystems with a closed N cycle (Fig. S3-S6).

In our data-model comparison, CLM5 was typically unable to capture inter-site variations in N recovery across PFT and fertilization levels, likely due to both model errors and measurement uncertainty (Fig. 2, Figs. S3-S6). For example, the measured total recovery of tracer at Harvard Forest evergreen forest appeared to increase with time (Nadelhoffer et al., 2004), which may be a result of changes in the sampling locations of soil cores between sampling events (i.e., soil sampling was done further inside plot boundaries in later years than in early sampling years). Thus, in Section 3.3, we discuss changes in the temporal patterns in simulated and observed N recovery averaged across sites, aggregated to the short-term (< 3 years) and longer-term (> 3 years) to capture the temporal break in slow and fast changes in simulated N recovery. The small number of sites for each forest type (deciduous or evergreen) and fertilizer treatment (ambient or fertilized) also limited statistical comparisons by these factors. Thus, to reach more statistically-robust conclusions, we combined both PFTs and fertilizer treatments in Section 3.3, and examine qualitative differences by these factors in Section 3.4.

### 3.3. Mean response in CLM5

### 3.3.1 Change in N recovery in CLM5

Across all sites and treatments, there were 14 field measurements from 13 experiments reporting $^{15}$N tracer recovery within 3 years of the start of N additions (i.e., short-term) and 14 field measurements from nine experiments reporting $^{15}$N recovery after 3 years (i.e., longer-term; Table 1). On the short-term, CLM5 with a closed N cycle estimated more than twice the mean plant N recovery of tracer (46 ± 12%; mean ± 1 SD across simulated sites) than was measured in the field (18 ± 12%; mean ± 1 SD across field measurements; $p < 0.001$; Fig. 3a). Short-term tracer recovery in soil was modeled to be 40 ± 10%, compared to the observational mean of 54 ± 22% (Fig. 3c), and was not statistically different from observations. Tracer recovery in plants decreased over time in both CLM5 and the field studies. On the longer-term, the closed N cycle led to modeled plant N recoveries (23 ± 6%) that were closer to observations (13 ± 5%) than they were on the short-term (Fig. 3b), but were still roughly twice the observed values ($p < 0.001$; Fig. 3b). The modeled decrease in mean plant N recovery over time corresponded with an increase in mean soil N recovery (to 59 ± 15%) that was smaller than observations on the longer-term (69 ± 18%; $p < 0.05$; Fig. 3d). The model's initially high plant N recovery and its later increase in soil N recovery indicate that CLM5 estimates soils to become a dominant sink for N on the long-term, but this response is a result of an over-competitive plant pool that transfers recovered N to soils through turnover of plant litter. Similar to the model

configuration with a closed N cycle, CLM5 with an open N cycle overestimated short-term ($38 \pm 7\%$; $p < 0.001$) and long-term ($20 \pm 6\%$; $p < 0.01$) plant N recovery, and underestimated long-term soil N recovery ($p < 0.001$). However, CLM5 with an open N cycle underestimated ($45 \pm 15\%$) short-term soil N recovery ($39 \pm 14\%$) compared to observations ($p < 0.05$).

On average, CLM5 with a closed N cycle simulated that most added N remained in the ecosystem over both short and longer timescales (Figs. 3e and 3f). Within 3 years of simulated tracer or fertilizer addition, the mean whole-ecosystem recovery of N under a closed N cycle was $87 \pm 14\%$, which was higher than the observational mean of $72 \pm 23\%$ ($p < 0.05$). On the longer term, simulations indicate that these forests retained added N with minimal loss ($83 \pm 17\%$), which was similar to the observational mean of $82 \pm 16\%$. CLM5 with an open N cycle simulated short-term ecosystem recovery ($77 \pm 14\%$) that was similar to observations. However, an open N cycle in CLM5 led to a longer-term ecosystem recovery that was lower than observations ($65 \pm 16\%$, $p < 0.01$).

### 3.3.2 Change in C response to N additions in CLM5

To scale and compare the effect of plant and soil N recoveries on forest C sinks between CLM5 with a closed N cycle and field measurements, we estimated changes in plant, soil, and total C stocks (i.e., sum of plant and soil stocks) in response to N tracer or fertilizer additions—referred to as $(dC/dN)_{plant}$, $(dC/dN)_{soil}$, $(dC/dN)_{total}$, respectively (Fig. 4). For the model, annual dC/dN values were computed directly (i.e., "direct approach") as the difference between the total plant or total soil C stocks between the baseline and "tracer" (or fertilizer) simulation divided by the difference in the amount of cumulatively added N between the two simulations (Eqn. 2). To estimate the effect of field-measured [15]N recoveries on forest C pools, we used the scaling exercise presented by Nadelhoffer et al. (1999b) that we described in Eqn. 3, where for each experimental timepoint, dC/dN is estimated for foliage, wood, bark, fine roots, and coarse roots (when available), the O horizon, and the mineral soil using the measured [15]N recovery in each pool and the published field-measured values of C:N for that site's particular pool (Table 2, Eqn. 3). Because differences in model and field-estimates of dC/dN can occur from differences in total N recovery, distribution of recovered N across sub-pools, or C:N ratios of sub-pools, we also used Eqn. 3 to compute a second model-based dC/dN using the same estimates of field-based C:N ratios, except that bark is not modeled in CLM5. This second, indirect approach allows us to remove sub-pool C:N ratios as a confounding factor in estimates of dC/dN. We used these two methods to calculate model dC/dN in order to a) directly show the model's overall C response to N additions and b) account for substantial differences in modeled and field-based approximations of C:N ratios in plant and soil pools (Table 2). It should be noted that this scaling exercise depends on the accuracy of the C:N ratios measured in the field, and operates under the assumption that C:N ratios of plant tissue and soil horizons stay constant over time. Despite these limitations, this budgeting method allows us to roughly compare differences in ecosystem C response to N additions between CLM5 and field measurements.

Averaged across experiments, the short-term direct estimate of $(dC/dN)_{plant}$ in CLM5 under a closed N cycle (right-most white bar in Fig. 4a) was similar to the field-based estimate (left-most white bar in Fig. 4a), despite the greater than two-fold difference in plant N recovery between modeled and observed values

(Fig. 3a). Within three years of N additions, the direct $(dC/dN)_{plant}$ in CLM5 was $26 \pm 8$ g C g$^{-1}$ N, compared to the field estimate of $19 \pm 14$ g C g$^{-1}$ N (Fig. 4a). On the decadal timescale, direct $(dC/dN)_{plant}$ in CLM5 became higher than the observational estimate ($28 \pm 7$ and $18 \pm 7$ g C g$^{-1}$ N, respectively, $p < 0.01$, Fig. 4b), though not as large as would be expected based on the amount of long-term plant N recovery. Differences between the directly-modeled $(dC/dN)_{plant}$ and field $(dC/dN)_{plant}$ may also be due to differences in C:N ratios of plant sub-pools. In particular, the C:N of wood in CLM5 is substantially lower (~266-293) than

field-based estimates ($411 \pm 110$, Table 2), compensating for the model's over-estimate in N recovery. When the same C:N ratios are used for both field- and model-based estimates of $(dC/dN)_{plant}$, the over-estimation of N recovery in CLM5 is carried more dramatically into $(dC/dN)_{plant}$: the simulated $(dC/dN)_{plant}$ (Fig 4a and 4b, center white bars) becomes substantially higher than observations on both the short ($52 \pm 15$ g C g$^{-1}$ N, $p < 0.001$) and the longer-term ($45$ g $\pm 15$ g C g$^{-1}$ N, $p < 0.001$).

        In soils, the direct approach to calculating $(dC/dN)_{soil}$ (right-most gray bars in Figs. 4a and 4b) in CLM5 with a closed N cycle estimated that soil C
stocks would decrease within three years of N additions ($-2 \pm 4$ C g$^{-1}$ N), and increase slightly in the long term ($5 \pm 3$ g C g$^{-1}$ N). This short-term decline in soil C in response to N additions is a result of decreasing litter C stocks. In contrast, observations indicate that soils retain $^{15}$N, which can be associated with C accumulation ($12 \pm 5$ g C g$^{-1}$ N; Fig. 4a, $p < 0.001$). On the long-term, measured soil C stocks ($15 \pm 5$ g C g$^{-1}$ N; Fig. 4b, $p < 0.001$) increased more than direct estimates from CLM5. When using the same soil C:N ratios to calculate $(dC/dN)_{soil}$ in CLM5 as estimated from observations, short- and longer-term soil $(dC/dN)_{soil}$ become similar between the model and field estimates (Fig 4a and 4b, center bars; $10 \pm 3$ g C g$^{-1}$ N and $15 \pm 6$ g C g$^{-1}$ N, respectively). Overall, CLM5
directly estimates a short-term $(dC/dN)_{total}$ of $24 \pm 7$ g C g$^{-1}$ N and a longer-term $(dC/dN)_{total}$ of $33 \pm 9$ g C g$^{-1}$ N, while the longer-term estimate ranges from 30 to 106 g C g$^{-1}$ N when using field-estimated C:N ratios. Generally, CLM5 with an open N cycle followed similar patterns as the model configuration with the closed N cycle for short and long-term plant and soil dC/dN, except that the indirect calculation of long-term soil dC/dN was statistically different from observations. Because of existing model limitations in N cycle representation, model-estimated values of dC/dN are intended to provide a sensitivity test of how the modeling of N fates can affect model estimates of ecosystem C response to N additions relative to what is expected from field measurements.

**3.4 Impacts of forest type and fertilization**

        In the field, forest types might respond to N deposition differently because of differences in their plant and ecosystem traits (Cornelissen, 1996). In CLM5, the evergreen and deciduous PFTs differ especially in their foliage C:N (see Table 2) and timing of plant N demand, which should alter decomposition and N mineralization. However, statistical comparisons between modeled and measured recoveries of N additions by forest type and fertilizer treatment are difficult to construct because of the small number of sites available for each category. Despite these limitations, we identified a few recurring differences between
a) deciduous and evergreen forests, as well as b) between ambient N deposition and fertilizer conditions.

Under ambient conditions, simulations using a closed N cycle had plants with notably more mean recovery of added N in two BDT (64%) forests than in four NET forests (44%) on the short term (Table 4, Fig. 5). On the long term, the amount of N recovered in modeled plants decreased in both forest types with no difference between the two PFTs (20-23%; Table 4, Figure 5). Conversely, simulated recovery of added N in soil was higher in NET (52%, n=4) than in BDT (32%, n=2) forests on the short term. However, CLM5 estimated similar long-term recoveries of N additions in soil (72 to 73%) in both PFTs, similar to long-term patterns of simulated plant N recovery. In contrast, measured recoveries of [15]N did not differ by forest type for plants or soils at either time point, except for short-term soil N recovery (Table 4, Figs. 5a and 5c). Simulations of CLM5 with an open N cycle followed similar patterns, except there was an underestimation of long-term soil recovery in both PFTs (Table 4).

Fertilization altered the simulated partitioning of N between plants and soils over short and long timescales, and generally reduced overall recovery of N in the ecosystem (Table 4, Figure 5). Simulations with a closed N cycle yielded lower recovery of N in deciduous plants (32%) than the two unfertilized stands (64%) on the short term. For the evergreen stands, modeled plant recovery of N did not differ between the four unfertilized and six fertilized stands, although there was considerable variation among the latter. Simulations for both forest types contradict observations showing that fertilization increases short-term plant recovery of tracer N regardless of forest type (Table 4). However, both measurements and simulations using a closed N cycle demonstrated that fertilization led to a decline in the amount of tracer retained in soil in the long-term (Table 4). The model configuration with an open N cycle generally followed similar trends as CLM5 with a closed N cycle.

Overall, we find evidence suggesting that a) under ambient treatments, CLM5 simulates differences in short-term plant N recovery between deciduous and evergreen forests, while measurements show no discernable difference (Table 4); b) the movement of N from plant to soil pools over the decadal timescale occurs in a distinctly different manner between CLM5 and measurements (Fig. 5); and c) the model and measurements respond differently to fertilizer in the short term, but both estimate declines in soil N recovery after at least 3 years of fertilizer additions (Table 4).

**4 Discussion**

This study compares estimates of ecosystem recovery of N deposition between CLM5—a land model with coupled C and N cycles—and long-term [15]N tracer experiments in temperate deciduous and evergreen forests. We examined CLM5, with a focus on simulations with a more closed N cycle, along three important axes of terrestrial C-N modeling: its ability to simulate a) the decadal patterns of N recovery in plant and soil pools, b) the plant and soil C responses to the model's estimates of N recovery, and c) the potential impacts of forest type and increases in N deposition on the partitioning of recovered N in ecosystems. Below, we also discuss the role of total N inputs and losses in ecosystem N recovery. Based on N recovery patterns, we identify some potential causes for the discrepancy between modeled and observed N fates in plants and soils—focusing on plant uptake and soil immobilization processes and recommending changes

for modeling plant-soil-microbial competition in future versions of CLM. We then compare our model estimates of the effects of N deposition on forest C sinks with other measurements in the literature, and discuss potential mechanisms behind differences in these responses. Last, we discuss the [15]N tracer dataset as a tool for evaluating CLM5 and other land models.

**4.1 Modeling Ecosystem Inputs and Losses**

Our analysis of CLM5 configured with an open N cycle (i.e, the default configuration of N cycling in CLM5) identified that the model continues to have large biases in N losses (Fig. 1)—similar to assessments of previous versions of the model (Koven et al., 2013; Thomas et al., 2013b; Houlton et al., 2015; Nevison et al., 2016). Specifically, CLM5 has unrealistically high rates of denitrification and low rates of N leaching and runoff compared to field measurements. We also identified that rates of pre-industrial N deposition in the input datasets were higher than expected from reconstructions for the Northeast United States and parts of Europe (Fig. 1; Fakhraei et al., 2016; Holland et al., 1999). Although simulations with an open N cycle shared some similar responses to N additions

as simulations with a closed N cycle, having higher N deposition and denitrification fluxes (i.e., the default version of CLM5) typically led to a) less total ecosystem recovery of N than simulations using a closed N cycle and b) an underestimation of long-term soil N recovery compared to observations. In adjusting N inputs and losses in CLM5 to better match field expectations, many of the simulated ecosystem stocks and fluxes (i.e., plant N, soil N, plant C, leaf area, and ANPP) remained similar to observations and simulations using an open N cycle (Table 3). Given that the openness of an ecosystem's N cycle changes the ecosystem's recovery of N inputs within a decadal timescale, we suggest that future model development not only test new mechanistic representations of N

fixation, losses from denitrification, nitrification, leaching, and runoff, but do so in concert with modified N deposition datasets to ensure that both inputs and losses capture field expectations.

**4.2 Plant-Soil N Competition, Plant N Uptake, and Soil N Immobilization**

Compared to observations, plants in CLM5 are a larger than expected short-term sink for N additions, with soils becoming the dominant sink for N thereafter, as leaf and fine root litter is incorporated into soils. Field experiments demonstrate the opposite pattern—large amounts of N are directly recovered in

soils from the start of tracer application, without passing through plants (Emmett et al., 1998a; Gundersen et al., 1998; Tietema et al., 1998; Nadelhoffer et al., 2004; Goodale, 2017). Two decades ago, Nadelhoffer et al. (1999b) used earlier [15]N tracer studies to illustrate similar problems in an earlier generation of models, but issues continue to persist in how these C-N competition processes are represented. The overly strong plant sink for N deposition in CLM5—both when the N cycle is open and closed—likely results in part from how the model handles N competition between plants and soil immobilization, as well as the model's representation of the plant uptake and soil immobilization processes themselves. Our results suggest the need for additional improvements to CLM5's

partitioning of N among plants, soils, and N loss pathways—similar to results shown in earlier studies with CLM4 (Thomas et al., 2013b)—even after substantial changes to the model's soil (Koven et al., 2013) and plant (Lawrence et al., 2018) C-N biogeochemistry have been made since earlier versions.

In CLM5, the amount of N that plants can acquire depends on how much inorganic soil N is available as well as the total demand for N from all modeled ecosystem processes, including soil immobilization, denitrification, and nitrification (Lawrence et al., 2018). When there is not enough N to meet the total demand from both plants and immobilization, inorganic soil N is divided between plants and soils by proportionately scaling their individual demands to the total demand. Plants can then take up their allocated portion of soil inorganic N if they have enough available C to pay for the cost of taking up that N. Given that plants recover too much added N on the short-term, regardless if the N cycle is open or closed, an option for reducing plant access to N without reducing the availability of inorganic N for immobilization is to increase the costs for plants to acquire N. To date, CLM has also used the long-standing assumption that plants acquire N only from inorganic N pools rather than organic N, and that plant demand does not affect N mineralization rates. However, evolving views of plant-soil interactions suggest more complex representations of both processes may be needed, in which plant mycorrhizal associations and priming can enable plants to acquire N from litter and soil organic matter, rather than relying solely on inorganic N (Schimel and Bennett, 2004; Phillips et al., 2013; Tang and Riley, 2014; Terrer et al., 2016; Zhu et al., 2016a; Sulman et al., 2017). Allowing plants to access N from organic as well as inorganic N pools in the model might seem a counterintuitive suggestion, given that plants already show excessive acquisition of newly added inorganic N. But the inclusion of these microbial-driven processes of N acquisition would both better match current understanding of plant-soil-microbial interactions, and could allow plants to meet their overall N demand even if competition for inorganic N by immobilization were increased (see below), and allow for added inorganic N to be retained in the soils longer, similar to observations.

Contrary to our model results, field experiments summarized here and elsewhere (e.g., Tietema et al. 1998, Curtis et al. 2011, Templer et al. 2012) demonstrate that soils dominate the fate of added $^{15}$N, and that this soil N sink is both rapid and direct, without passing through plants. For example, $^{15}$N tracer studies at two sites simulated here, Alptal (Providoli et al., 2006) and Arnot Forest (Goodale et al., 2015), as well as at other temperate forests (Seely et al., 1998; Perakis and Hedin, 2001; Hagedorn et al., 2005; Lewis and Kaye, 2012), show that large quantities of $^{15}$N can be recovered in association with soil organic matter pools within days to weeks of its addition, including in the soil clay or "heavy" fractions, which are generally the most stable components of soil. CLM5 currently immobilizes little N directly into soil, particularly when N input fluxes are high (open v. closed N cycle, and fertilized v. ambient simulations). However, it should be noted when comparing model estimates and observations to each other that plant access to existing soil N and N additions in the field is not represented identically in CLM5. For example, N additions (e.g., N deposition) in CLM5 are directly added to the dissolved inorganic N pool, which the model immediately distributes throughout the soil column according to an exponential profile. In contrast, the $^{15}$N tracer in the field is typically applied on top of the leaf litter layer. Although the model and field experiments differ in how they apply the N tracer to soil (directly into the inorganic soil N pool versus to the

top of litter, respectively), the large magnitude of the observed soil N sink and the model's poor ability to reproduce it suggests that modeling a stronger soil immobilization sink should be a priority.

Several soil processes and ecosystem traits that are involved in immobilization are not currently represented in CLM5, and could help increase the N demand for soil immobilization. The model's current soil C-N dynamics (Koven et al., 2013) were adapted largely from the CENTURY model, which has implicit microbial processes rather than explicit representation of microbial N uptake and turnover—processes that form a dominant pathway for N incorporation into soil organic matter (Bingham and Cotrufo, 2016). Incorporating an explicit representation of microbial biomass, and providing microbes with access to inorganic soil N before plants can access it should increase rates of soil immobilization of added inorganic N—particularly because microbial activity and demand for N is greatest at the soil surface (Iversen et al., 2011; Li and Fahey, 2013) where fresh C inputs are greatest, C:N ratios are high, and microbes have the opportunity to rapidly capture N deposition. Previous modeling work has shown that explicitly representing microbes improves soil C stock projections (Wieder et al., 2013), and that more precisely representing plant and microbial biomass and their enzyme affinity for inorganic N better captures the fates of N in grasslands (Zhu et al., 2017).

Field experiments also demonstrate that when ecosystem N additions increase (e.g., with fertilizer), the recovery of $^{15}$N subsequently increases in plants and decreases in soils on the short and longer-term (Nadelhoffer et al., 1999b; Templer et al., 2012). But under both open and closed N cycles, CLM5 estimated a different short-term plant response depending on forest type, generally estimating a decrease in plant N recovery in deciduous forests and a slight increase in evergreen forests (Table 4). This response is likely due in part to excess plant uptake of N in forests after the "tracer" is applied in the model. However, the model generally produced a decline in long-term soil N recovery in response to fertilizer treatment (Table 4), except for the mean soil N recovery in evergreen forests with open N cycles. Model development that incorporates plant-soil-microbial dynamics, as described above, would likely yield larger decreases in soil N recovery deposition and fertilizer because soils would recover more N additions under ambient conditions before fertilizer is added. It is important to note that additions of parameters or process-based representation of ecological processes can add uncertainty to model projections. To limit this added uncertainty, new model representations should be designed and evaluated using robust and representative, process-based datasets—as discussed in modeling papers, including Prentice et al. (2015), Lovenduski and Bonan (2017), Lombardozzi et al. (2018), and Sulman et al. (2018).

**4.3 Forest Carbon Sequestration from N Deposition**

Despite plants in CLM5 being overly competitive for ecosystem N additions, the model did not dramatically overestimate the response of ecosystem C pools to these additions in simulations with closed N cycles when compared to observations (Fig. 4a and 4b). Instead, CLM5 directly estimated a longer-term $(dC/dN)_{total}$, ranging from 19 to 45 g C g$^{-1}$ N in simulations with closed N cycles, close to the lower bound of the 50 to 75 g C g$^{-1}$ N range of dC/dN estimated from measurements of forest growth across a N deposition gradient (Sutton et al., 2008), and similar to the range measured in temperate forests (-2 to 48 kg C kg$^-$

[1] N) summarized by Frey et al. (2014). Similarly, CLM5's estimates of $(dC/dN)_{total}$ for simulations with closed N cycles also fall within the range modeled by O-CN (2 to 79 g C $g^{-1}$ N) and CLM4 (24 to 30 g C $g^{-1}$ N) for temperate forest ecosystems (Zaehle and Friend, 2010; Thomas et al., 2013a). However, modeled estimates of dC/dN can be difficult to interpret against field estimates of dC/dN or those reported by other models because estimates of dC/dN depend not only on N recovery in plant and soil pools, but also on the C:N ratios of these pools.

For example, our directly-measured model results for $(dC/dN)_{total}$ initially appeared counterintuitive because CLM5 estimated approximately twice as much recovery of N in plants than measured in field experiments. In CLM5, the C:N ratio of dead wood is approximately half the value of what is measured at field sites (Table 2). In addition, plant C:N ratios in the model appear to have an important role in immobilization—as a higher C:N ratio of plant litter in NET forests led to higher rates immobilization compared to BDT forests (Figs. 2a and 2b). When the difference in C:N ratios between the model and field measurements were accounted for by using the mean C:N ratios from available data at our sites, the model's estimated $(dC/dN)_{total}$ increased to a range of 30 to

106 g C $g^{-1}$ N, a range much larger than observed estimates. While this range overlaps with the observed ranges, the high end of the range exceeds the observations. This discrepancy between observed and modeled C:N ratios would also apply to the default version of CLM5, where ecosystems generally have an open N cycle. Thus, we recommend that options for improving the calculation of C:N ratios of plant pools in CLM5 be explored in order for CLM5 to more accurately and mechanistically model the correct ecosystem C responses to N additions. To accomplish this, additional field measurements would be needed to evaluate changes to model estimates of C:N ratios and to constrain them to reasonable values.

CLM5 simulations with both open and closed N cycles indicated that adding N to temperate forests yielded on average a small loss of soil C on the short term (due to declines in the litter pool), and a small increase on the long-term (Fig. 4). These dynamics are consistent with modeled relief of N limitation to litter decomposition in the short-term (Bonan et al., 2012), and with increased plant NPP on the long-term. Yet reviews of long-term N addition studies in mature forests show increases in soil C stocks that are associated with reduced rates of decomposition rather than an increase in plant litter production (Janssens et al., 2010; Frey et al., 2014). The increase in soil C stocks might be explained by the changes to the plant-soil-microbial feedbacks described above, in which plant

acquisition of N under ambient N availability is mediated by microbial symbionts that drive decomposition, and these processes slow when external N supplies increase. Explicit representation of plant-soil-microbial feedbacks, such as in Sulman et al. (2017), could improve model representation of both soil C responses to N addition as well as the plant and soil N fates discussed above.

**4.4 Evaluating N Fate in Modeled Forests**

      Model evaluation can often be a challenge because of the limitations in the availability and consistency of how field measurements are taken and

analyzed. To build a useful dataset that can be leveraged for model evaluation and for potential benchmarking (Luo et al., 2012; Collier et al., 2018), differences between N recovery estimates from different field experiments need to be reconciled. Even when focusing on temperate deciduous and evergreen forests, we

found a wide variation in the measurements of N recovery within each forest type. This range in measurements made it difficult to identify how much of the mismatch between measurements and CLM5 was a consequence of model weaknesses alone. For example, calculations of tracer recovery in soils can depend on uncertainties associated with soil organic N pool measurements—usually the largest and most difficult pool to quantify in terrestrial ecosystem (Nadelhoffer et al., 2004). Accuracy at some sites in part depends on how well-constrained measurements of bulk density are, the number of soil cores taken, and the depth of the soil cores—which ranged across sites from as shallow as 5 cm in Bear Brook Forest to as deep as 50 cm at Arnot Forest. In long-term field studies, changes to field sampling procedures over time add uncertainty to comparisons between observed and modeled temporal trends, as well as uncertainty in evaluating the model's N recovery response to fertilizer and forest type. Increasing the number of long-term [15]N tracer experiments at evergreen and deciduous forests could help constrain estimates of N recovery in plant and soil pools at these two forest types—leading to a more robust dataset for future model evaluations.

In addition, measurement data from other biomes are needed to evaluate the global impact of CLM5's estimates of N recovery in plant and soil pools. Currently, most tracer experiments are in North America and Europe, which represent a subset of ecosystem types and climates that exist globally. One of the few [15]N experiments in tropical forests indicates that soil retention of [15]N is similar to that found in temperate forests (Gurmesa et al., 2016), even though tropical forests are typically limited by phosphorus availability and N (Hedin et al., 2009). Additional tracer experiments in the tropics would allow us to evaluate whether this pattern is anomalous for forests in this biome and whether the response of CLM5's tropical PFTs to N additions is similar to that modeled for temperate PFTs. Finally, increasing the number of sampling events at current and future field sites will expand our capacity to test more nuanced hypotheses about temporal patterns in dC/dN over time, both during the first few years after tracer application when more dramatic changes in [15]N recovery in plant and soil pools occur, and for timescales longer than two decades.

**5 Conclusions**

The accuracy of Earth system model projections of land C storage relies on how well land models can simulate the long-term responses of plant and soil C stocks to environmental change, including to shifts in N deposition. To evaluate a land model commonly used in global model intercomparison projects, we simulated temperate and evergreen forests in CLM5 with open and closed N cycles, and subsequently compared the modeled fate and effect of N additions on C stocks against measurements from long-term [15]N tracer experiments. Overall, we found that a sizable portion of N additions in CLM5 is taken up first by plants and then moved into long-term soil pools through the recycling of plant litter—contrary to field experiments that indicate N deposition is predominately immobilized and retained in soils. Given that CLM5 overestimated plant N recovery on the short and longer-term, the modelled plant dC/dN responses were smaller than expected—although the model slightly over-estimated long-term plant dC/dN when a closed N cycle was used. A larger plant dC/dN did not typically occur because the model's C:N ratios for wood were smaller than those generally found in the field, which compensated for the model's plant pool that

was overly competitive for N additions. For similar reasons, CLM5 underestimated soil dC/dN, even though the modeled mean N recoveries in soil were similar to observations (except for when the model under-estimated long-term soil N recovery when using an open N cycle). Overall, our data-model comparison suggests further exploration into a) more accurate N input data, b) better representations of N fixation, denitrification, nitrification, leaching, runoff, and C:N ratios in CLM5, and c) incorporating additional plant-soil-microbial processes in land models to increase soil immobilization and reduce initial plant N uptake of

N additions.

**Data availability**

Model output and scripts used to analyze data presented in this manuscript are available upon request. Modeled N recoveries and $^{15}$N tracer data are available in Table S1 and at https://doi.org/10.5281/zenodo.2772160. The most recent version of CLM5 is publicly available through the Community Terrestrial System Model (CTSM) git repository (https://github.com/ESCOMP/ctsm).

**6 Acknowledgements**

We would like to acknowledge high-performance computing support from Cheyenne (doi:10.5065/D6RX99HX) and Yellowstone (ark:/85065/d7wd3xhc)

provided by the National Center for Atmospheric Research's Computational and Information Systems Laboratory, sponsored by the National Science Foundation and other agencies. SJC, WRW, DLL, CLG and RQT were supported in part by the US Department of Agriculture NIFA Award 2015-67003-23485. SJC, PGH, and CLG also received support from NSF-ETBC 10-21613. WRW was also supported by the Environmental Protection Agency's National Center for Environmental Assessment, through an Interagency Agreement with the National Science Foundation and the National Center for Atmospheric Research DW-49-92447301-0. PGU was supported by Aage V. Jensen Naturfond. KJN acknowledges sabbatical support from the University of Copenhagen. We are especially

grateful to Erik Kluzek, Keith Oleson, Richard Valent, Silverio Vasquez, Davide Del Vento, and other staff at NCAR's Computational and Information Systems Laboratory for their support with running simulations. We are also grateful for the thoughtful comments and suggestions from our *Biogeosciences* reviewers and editor.

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

**Table 1: Site information for long-term $^{15}$N experiments in Europe and North America, including corresponding plant functional type (PFT) in CLM5, which is either broadleaf deciduous temperate (BDT) or needleleaf evergreen temperate (NET) tree. The year of stand establishment is the year in which we implemented a harvest in the model to simulate the forest's reported stand age. N deposition is reported as throughfall or the sum of wet and**

5 **dry N deposition as compiled from available literature. Tracer experiments at plots receiving ambient N deposition and fertilizer treatments are described on separate lines for each site. Please refer to Section 2.4 for details on how these field experiments were simulated using CLM5.**

| Site | Latitude, Longitude | PFT | Year of Stand Establishment | N Deposition (g N m$^{-2}$ y$^{-1}$) | Years N Fertilizer Applied at Field Site | Amount of N Fertilizer Applied at Field Site (g N m$^{-2}$ y$^{-1}$) | First Year Tracer Applied | Number of Years After Tracer Application that Recovery was Measured |
|---|---|---|---|---|---|---|---|---|
| Harvard (USA) | 42°30' N, 72°10' W | BDT | 1945[a] | 0.8[a] | None | 0.0[b] | 1991[b] | 1, 8, 17 |
| | | | | | 1988-Present[b] | 5.0[b] | 1991[b] | 1, 8, 17 |
| Arnot (USA) | 42°17'N, 76°38'W | BDT | 1911[c] | 0.9[c] | None | 0.0[f] | 2007[c] | 1, 6 |
| Bear Brook (USA) | 44°52'N, 68°06W | BDT | 1945[d] | 0.8[d] | 1989-2016[e,f] | 2.5 | 1991* | 1 |
| Harvard (USA) | 42°30' N, 72°10' W | NET | 1926[a] | 0.8[a] | None | 0.0 | 1991[b] | 1, 8, 17 |
| | | | | | 1988-Present[b] | 5.0[b] | 1991[b] | 1, 8, 17 |
| Klosterhede (Denmark) | 56°29'N, 8°24'E | NET | ~1920[h] | 2.0[h] | None | 0.0 | 1992[g] | 1, 17 |
| | | | | | 1992-1996 and 1999-Present[h] | 3.5[h] | 1992[g] | 1, 17 |
| Gårdsjön (Sweden) | 58°04'N, 12°03'E | NET | 1910[i] | 1.5[j] | 1991-Present[i] | 4.0[i] | 1992[k] | 2 |
| Aber (Wales) | 53°13'N, 4°00'W | NET | 1960[g] | 1.5[g] | 1990-Present | 3.5[g] | 1992[g] | 3 |
| | | | | | 1990-Present | 7.5[g] | 1992[g] | 3 |
| Alptal | 47°02, | NET | ~ 1750[l] | 1.7[l] | None | 0.0 | 2000 | 1, 3, 9 |

| | | | | | | | |
|---|---|---|---|---|---|---|---|
| (Switzerland) | 8°43'E | | | | 1995-Present[l] | 2.5[l] | 2000 | 2, 7, 14 |

[a]Magill et al. (2004), [b]Nadelhoffer et al. (2004), [c]Goodale (2017), [d]Elvir et al. (2006), [e]I. Fernandez, pers. comm., [f]Nadelhoffer et al. (1999a), [g]Tietema et al. (1998), [h]Gundersen (1998), [i]Seftigen et al. (2013), [j]Moldan et al. (2006), [k]Kjønaas and Wright (2007), [l]Krause et al. (2012), *we did not simulate the second tracer application that took place in 2012 at the fertilized plots in Bear Brook.

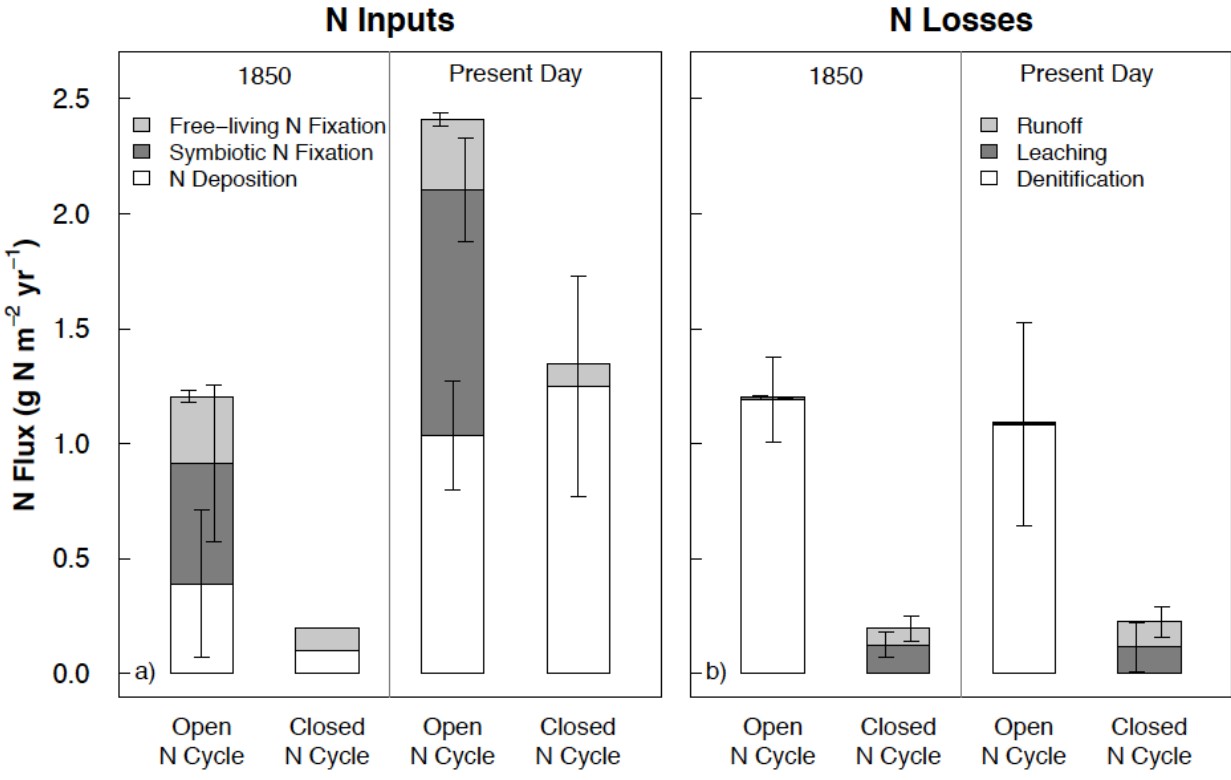

**Figure 1: Rates of a) nitrogen (N) inputs and b) N losses averaged across sites for 1850 (using the last 20 years of the spin-up simulation) and the present day (using the last 20 years of the historical baseline simulation; see Section 2.3) for ecosystems with open and closed N cycles. Mean inputs and losses include sources that were > 0.1% of total fluxes; N deposition is based on input datafiles while other fluxes are modelled in CLM5 (see Section 2.2). For ecosystems using a closed N cycle, N deposition, free-living N fixation, and N deposition were set to observation-based estimates; denitrification was turned off because of the uncertainty around the portion of N losses due to this loss term (See Section 2.3.1). Error bars represent 1 standard deviation simulated across sites.**

**Table 2: Mean C:N values for plant and soil pools reported in site-specific literature and simulated by CLM5 (averaged across sites). Model means (± 1 standard deviation across simulated sites) are the average of C:N ratios from the last 20 years of the baseline simulation from simulations with a closed N cycle. Observational means (± 1 standard deviation across measured site data) are based off site-reported or field-estimated values for C:N ratios, which are listed for each site, along with their references (which describe the sampling methods for each pool), in Table S3.**

| C:N | Observational Estimate | Model BDT Mean[a] | Model NET Mean[a] |
|---|---|---|---|
| Leaf | 37 ± 12 | 24 ± 0.4 | 63 ± 0.9 |
| Fine Roots | 45 ± 10 | 43 ± 0.7 | 46 ± 0.7 |
| Coarse Roots | 90 ± 20 | 26 ± 0.5 (live)[a] <br> 266 ± 7 (dead)[a] | 28 ± 0.3 (live)[a] <br> 266 ± 7 (dead)[a] |
| Wood | 411 ± 110 | 26 ± 0.5 (live)[a] <br> 266 ± 7 (dead)[a] | 26 ± 0.5 (live)[a] <br> 293 ± 8 (dead)[a] |
| Bark | 182 ± 55 | Not modeled | Not modeled |
| Organic Layer | 26 ± 6 | 59 ± 6 | 99 ± 10 |
| Mineral Layer | 21 ± 6 | 11 ± 0.004 | 11 ± 0.006 |

[a]In CLM5, wood and coarse roots have the same C:N ratios and are split into live and dead pools. In this table, we have listed modelled live biomass as coarse roots and modelled dead biomass as wood.

**Table 3: Comparison of mean nitrogen (N) and carbon (C) stocks and annual fluxes in modelled ecosystems with an open and closed N cycle at the end of baseline historical simulations. Sites fall into one of two plant functional types (PFT) in CLM5, broadleaf deciduous temperate (BDT) or needleleaf evergreen temperate (NET) tree. Model values are the mean of the last 20 years of the historical simulation and include total plant (above and belowground plant pools) and soil (excluding coarse woody debris and to the depth of field measurements) stocks. The total depth of the soil column in CLM5 is approximately 7.5 m at our simulated sites. ANPP and C stocks reported in the literature as organic matter or biomass were converted to units of C by assuming that 50% of organic matter is C.**

| Site (PFT) | Data | Aboveground Net Primary Productivity (ANPP) $(g\ C\ m^{-2}\ y^{-1})$ | Maximum Annual Leaf Area Index (LAI) $(m^2\ m^{-2})$ | Plant C $(g\ C\ m^{-2})$ | Soil C $(g\ C\ m^{-2})$ | Plant N $(g\ N\ m^{-2})$ | Soil N $(g\ N\ m^{-2})$ |
|---|---|---|---|---|---|---|---|
| Harvard (BDT) | Open N cycle | 271 | 2.9 | 6820 | 3060 (20 cm) 5930 (total) | 40 | 269 (20 cm) 527 (total) |
| | Closed N cycle | 260 | 2.7 | 7010 | 3310 (20 cm) 6470 (total) | 39 | 292 (20 cm) 576 (total) |
| | Observation | 373[a] | 5.3[b] | 10,110[c] | 9710[c] | 24[d] | 415 (20 cm)[d] |
| Bear Brook (BDT) | Open N cycle | 260 | 2.7 | 7720 | 1540 (10 cm) 7330 (total) | 42 | 133 (10 cm) 650 (total) |
| | Closed N cycle | 281 | 3.0 | 8750 | 1800 (10 cm) 8720 (total) | 48 | 155 (10 cm) 775 (total) |
| | Observation | 446[e] | 7.6[f] | 5830[g] | 5910[g] | 20[g] | 285 (10 cm)[g] |
| Arnot (BDT) | Open N cycle | 279 | 3.0 | 10680 | 8510 (50 cm) 10720 (total) | 56 | 755 (50 cm) 954 (total) |
| | Closed N cycle | 270 | 2.9 | 10930 | 9130 (50 cm) 11540 (total) | 56 | 810 (50 cm) 1028 (total) |
| | Observation | 270[h] | NR | 10380 | 7270[h] | 37[h] | 645 (50 cm)[h] |
| Harvard (NET) | Open N cycle | 339 | 4.0 | 10690 | 3610 (20 cm) 7010 (total) | 53 | 305 (20 cm) 610 (total) |
| | Closed N cycle | 287 | 3.3 | 9880 | 3610 (20 cm) 7120 (total) | 47 | 305 (20 cm) 621 (total) |
| | Observation | 294[a] | 4.4[b] | 12370[c] | 11050[c] | 21[d] | 460 (20 cm)[d] |
| Gårdsjön (NET) | Open N cycle | 398 | 4.9 | 14580 | 4980 (30 cm) 9750 (total) | 69 | 539 (38 cm) 850 (total) |
| | Closed N cycle | 422 | 5.3 | 15830 | 5520 (30 cm) 10920 (total) | 75 | 606 (38 cm) 958 (total) |
| | Observation | 275[i] | NR | NR | 18880[j] | 82[i] | 584 (38 cm)[j] |
| Aber (NET) | Open N cycle | 388 | 4.8 | 9130 | 4760 (30 cm) 8940 (total) | 49 | 417 (30 cm) 793 (total) |
| | Closed N cycle | 348 | 4.2 | 8320 | 3760 (30 cm) 6920 (total) | 43 | 329 (30 cm) 613 (total) |
| | Observation | NR | NR | NR | 17700 (30 cm)[k] | NR | 955 (30 cm)[l] |
| Klosterhede (NET) | Open N cycle | 366 | 4.4 | 13130 | 4030 (30 cm) 7740 (total) | 61 | 339 (30 cm) 670 (total) |
| | Closed N cycle | 413 | 5.2 | 14520 | 4170 (30 cm) 7990 (total) | 68 | 359 (30 cm) 700 (total) |
| | Observation | 352[k] | 6.0[m] | 12450[k] | 13270 (30 cm)[k] | 91[k] | 441 (30 cm[k] |
| Alptal (NET) | Open N cycle | 421 | 5.2 | 17450 | 7740 (25 cm) 14160 (total) | 80 | 657 (25 cm) 1234 (total) |
| | Closed N cycle | 382 | 4.6[a] | 14420 | 6310 (25 cm) 11570 (total) | 66 | 542 (25 cm) 1014 (total) |
| | Observation | 355[n] | 3.8[n] | 13140[o] | 9810[o] | 60[n] | 435 (25 cm)[n] |

NR: not reported in literature
[a]Magill et al. (2004)
[b]Zhao et al. (2011)
[c]Frey et al. (2014)

[d]Nadelhoffer et al. (2004)
[e]Magill et al. (1996)
[f]Elvir et al. (2006)
[g]Nadelhoffer et al. (1999a)
[h]Goodale (2017)
[i]Kjønaas and Stuanes (2008)
[j]Emmett et al. (1998b)
[k]personal communication with P. Gundersen
[l]Emmett et al. (1998b)
[m]Beier (1998)
[n]Krause et al. (2012)
[o]personal communication with P. Schleppi

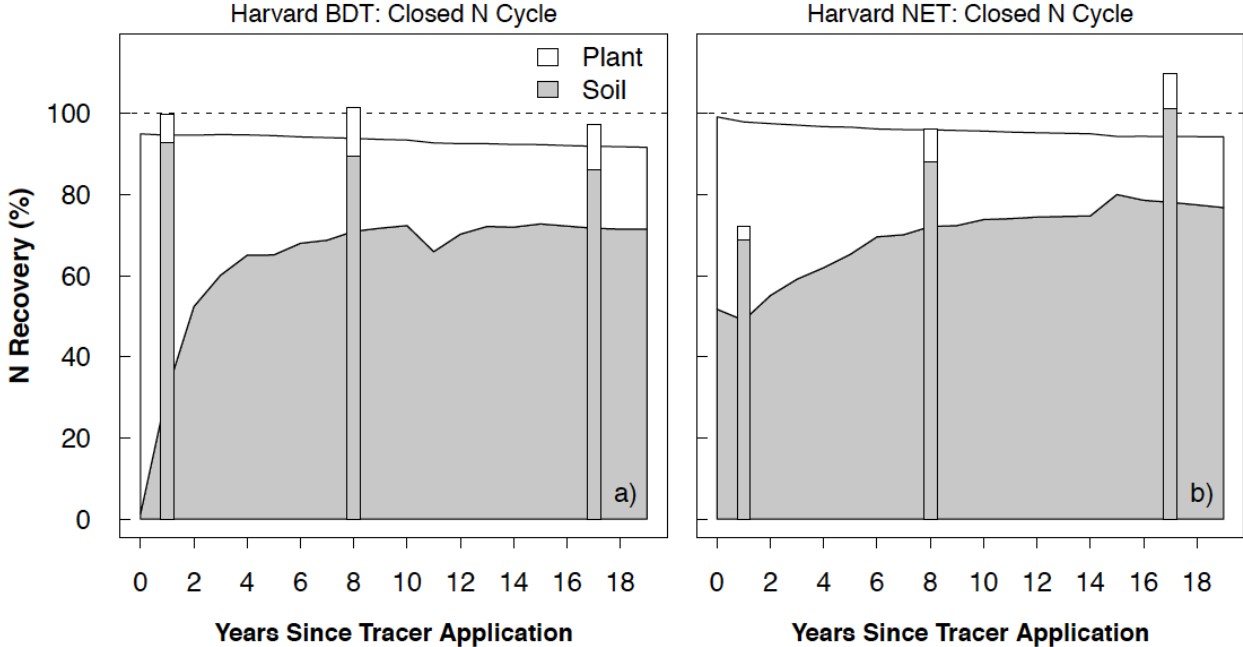

**Figure 2: Modeled recovery of N additions for Harvard Forest broadleaf deciduous temperate tree (BDT; left panel) and needleleaf evergreen temperate tree (NET; right panel). Simulations were done using a closed N cycle and are compared to observations (stacked bars). The modeled soil stock includes the organic soil, inorganic soil, and litter pools (excluding coarse woody debris). Recovery is calculated as the difference in stock size between a control simulation and a simulation with a "tracer" added as 0.5 g m$^{-2}$ between April-September in year 0. Plots of recoveries at all other sites and for the version of CLM5 with an open N cycle at Harvard Forest are shown in Figs. S3-S5.**

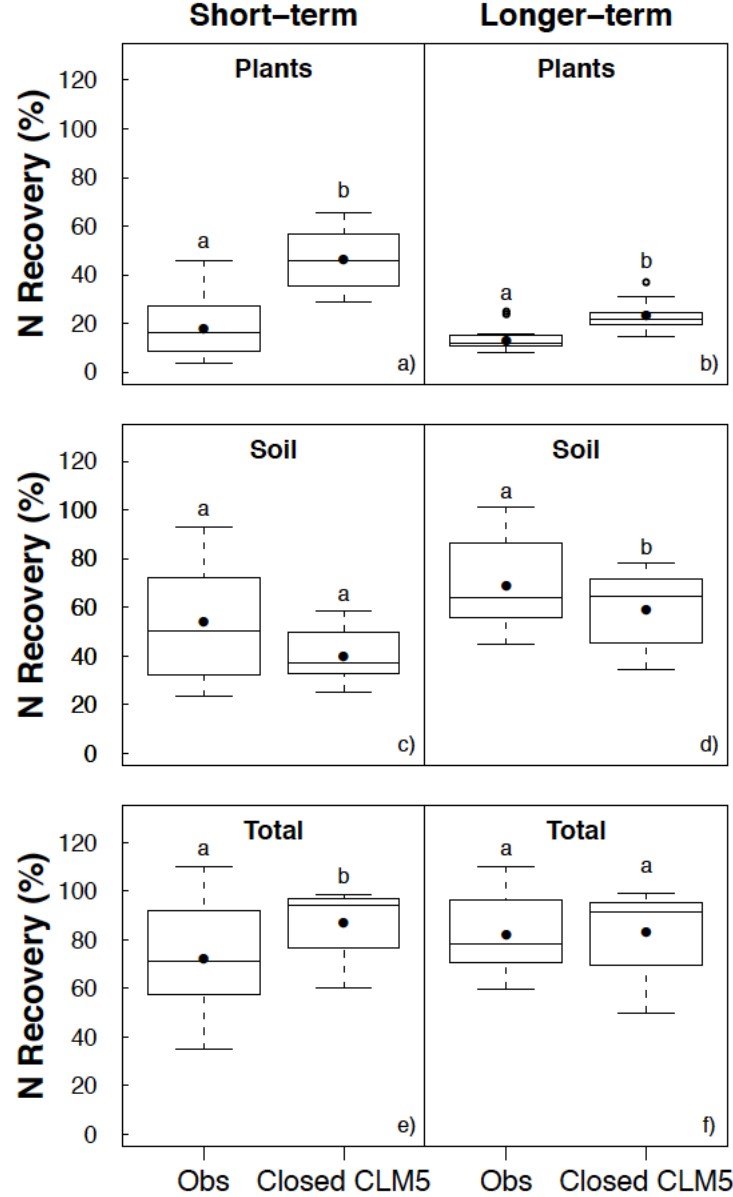

**Figure 3: Boxplot showing the mean (filled dot), median (horizontal line), 1st quartile, and 3rd quartile of N recovery (%) on the short-term (< 3 years) and longer-term (> 3 years) from ¹⁵N experiments with available data (Obs) and model simulations using a closed N cycle (Closed CLM5). Field and modelled data are an aggregate of values from both plant functional types (PFTs) and all N fertilization or ambient conditions. Whiskers extend to the minimum and maximum N recoveries that are not outliers, which are represented by open circles. Variation in observations occurs from differences measured across sites; variation in the model data occurs from differences estimated by CLM5 across modelled sites. Different letters indicate groups that are statistically different (*p* < 0.05).**

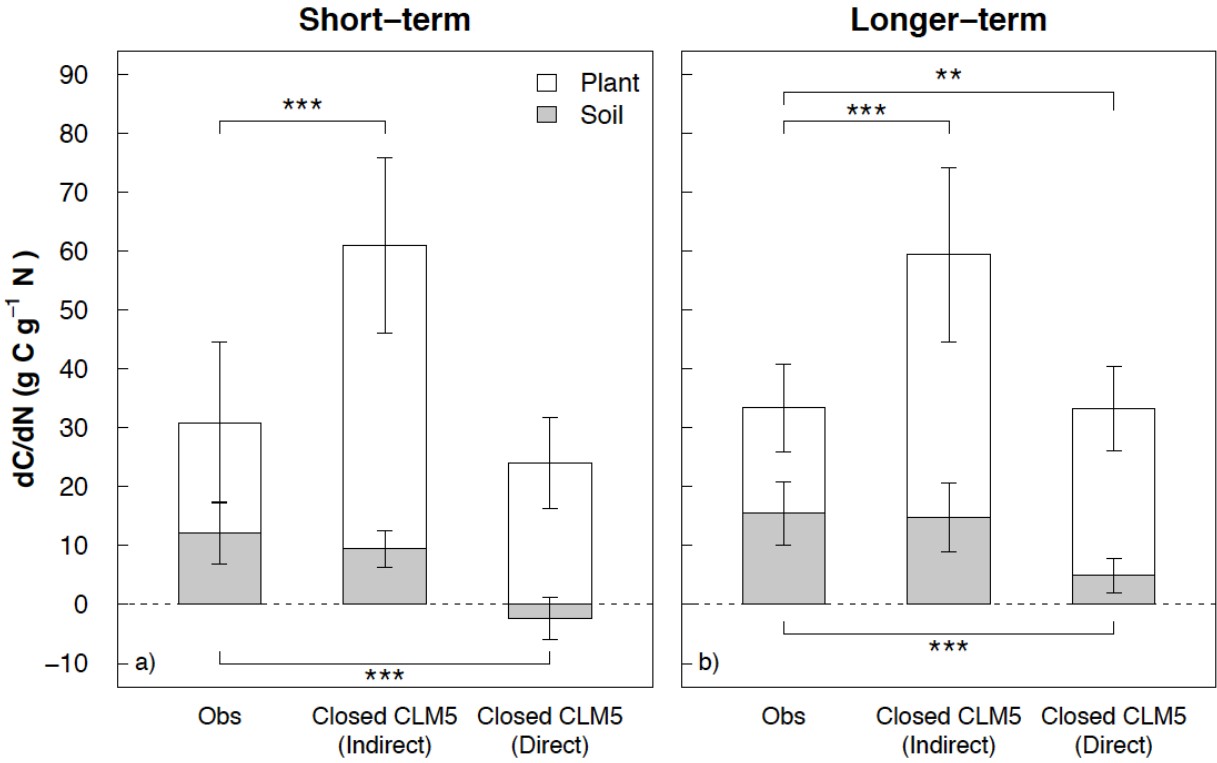

**Figure 4: Mean short-term (< 3 years) and long-term (> 3 years) response of C stocks to added N (dC/dN) estimated from $^{15}$N experiments (Obs) and model simulations using a closed N cycle (Closed CLM5). The dC/dN for field experiments (left-most bars in Fig. 4a and 4b) is calculated by multiplying the site-level $^{15}$N recovery for foliage, wood, bark, fine roots, coarse roots, O horizon, and mineral soil horizon by the C:N ratio for each pool (Eqn. 3, see Table S3 for C:N ratios) and then averaging the total plant or total soil dC/dN across sites. For equivalent comparison to observations, model dC/dN (center bars in Figs. 4a and 4b) is calculated indirectly using observational C:N ratios (Indirect) and Eqn 3 as well as directly from the model using the model's C:N ratios (Direct) according to Eqn 2 (right-most bars in Figs. 4a and 4b). For plant dC/dN, data from Gårdsjön, Aber low, and Aber high experiments were not included because sub-pools from plants were not reported. For soil dC/dN, data from Gårdsjön and Bear Brook Fertilized were not included because sub-pools from soils were not reported. Error bars represent 1 standard deviation. Variation in the observations occurs from differences in N recoveries and C:N ratios measured across plant and soil sub-pools and sites; variation in the indirect model estimates occurs from differences in N recoveries estimated by CLM5 across modelled sites and from variation in measured C:N ratios; variation in the direct model estimates occurs from differences in CLM5's stocks and C:N ratios across modelled sites. Statistical differences between observations and model-estimates of dC/dN are indicated with asterisks for plants (above the white bars) and for soils (below the gray bars), where \*\* represents $p < 0.01$, and \*\*\* represents $p < 0.001$.**

**Table 4: N recovery in CLM5 simulations using an open N cycle (i.e., the default version of CLM5) and a closed N cycle compared to observations for plant and soil stocks on the short (< 3 years) and longer-term (> 3 years). Data are separated according to PFT (broadleaf deciduous temperate; BDT and needleleaf evergreen temperate; NET trees) and fertilization treatment, as well as aggregated across all sites (combined across PFTs and fertilization treatment). The number of data points for each N recovery is listed in parentheses.**

| Timescale and Stock | Data | BDT | | NET | | Combined |
| --- | --- | --- | --- | --- | --- | --- |
| | | Ambient (%) | Fertilized (%) | Ambient (%) | Fertilized (%) | Combined (%) |
| Short-term Plant | Open N cycle | 49 (2) | 36 (2) | 35 (4) | 37 (6) | 38 (14) |
| | Closed N cycle | 64 (2) | 32 (2) | 44 (4) | 47 (6) | 46 (14) |
| | Observations | 9 (2) | 18 (2) | 13 (4) | 26 (6) | 18 (14) |
| Short-term Soil | Open N cycle | 25 (2) | 26 (2) | 54 (4) | 37 (6) | 39 (14) |
| | Closed N cycle | 32 (2) | 34 (2) | 52 (4) | 37 (6) | 40 (14) |
| | Observations | 76 (2) | 48 (2) | 49 (4) | 52 (6) | 54 (14) |
| Long-term Plant | Open N cycle | 15 (3) | 18 (2) | 20 (4) | 23 (5) | 20 (14) |
| | Closed N cycle | 23 (3) | 17 (2) | 20 (4) | 28 (5) | 23 (14) |
| | Observations | 10 (3) | 11 (2) | 13 (4) | 16 (5) | 13 (14) |
| Long-term Soil | Open N cycle | 55 (3) | 34 (2) | 43 (4) | 44 (5) | 45 (14) |
| | Closed N cycle | 72 (3) | 35 (2) | 73 (4) | 50 (5) | 59 (14) |
| | Observations | 79 (3) | 60 (2) | 78 (4) | 58 (5) | 69 (14) |

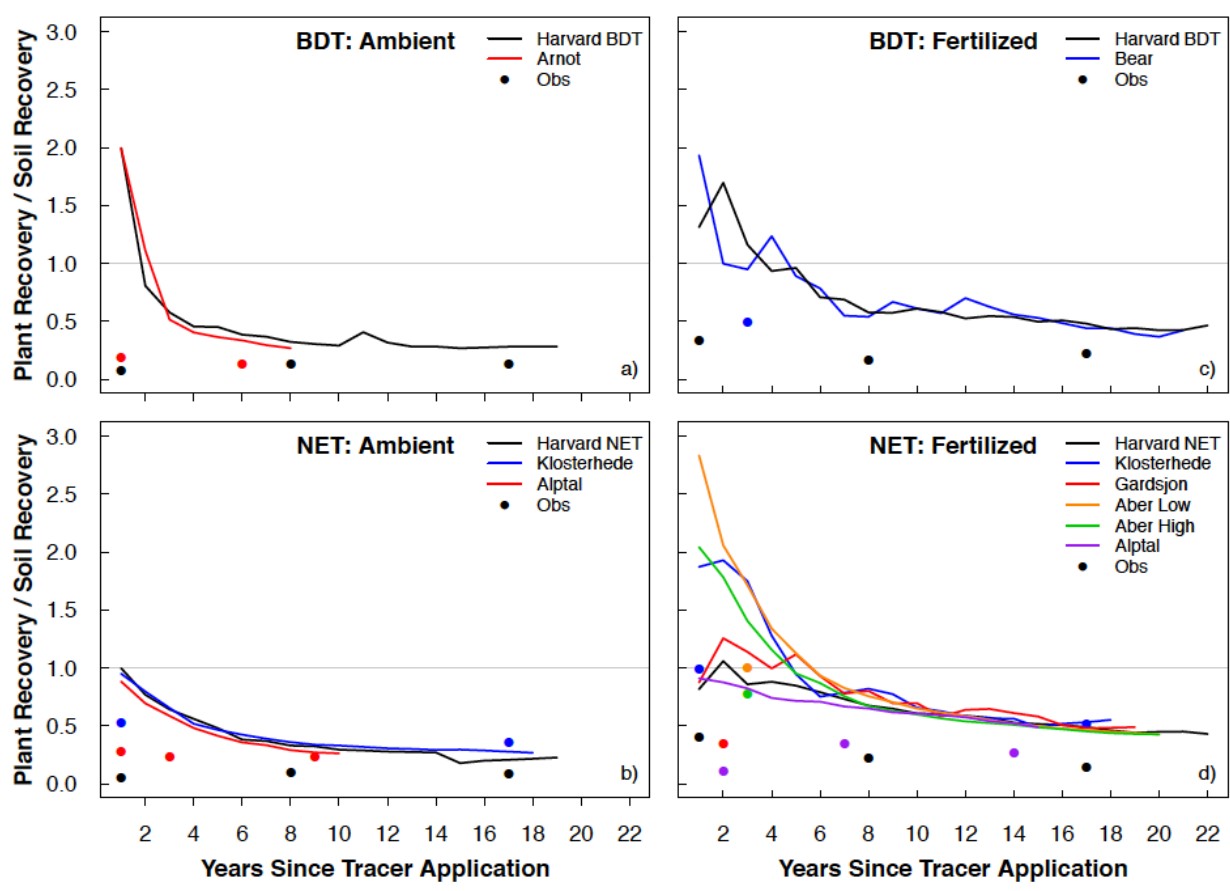

**Figure 5: Ratio of plant recovery and soil recovery of added N into a,c) broadleaf deciduous temperate (BDT) and b,d) needleleaf evergreen temperate (NET) forests for sites with a closed N cycle under ambient deposition (a,b) and fertilized (c,d) conditions. Circles represent the ratio of plant to soil recovery of $^{15}$N as measured in field experiments. A ratio of 1.0 represents equal recovery of N in plant and soil pools.**