# Peer review of "Decadal fates and impacts of nitrogen additions on temperate forest carbon storage: A data-model comparison"

_Biogeosciences, 2018_

## Referee Comment (RC1) · Anonymous Referee #1 · 3 Jan 2019

Cheng et al. present a model-data comparison of ecosystem N recovery at several temperate forest sites, using two versions of the CLM5 land model and a number of 15N tracer field experiments. They also use the results to give estimates of ecosystem C storage responses to changes in N availability. The paper is very well written and thoroughly describes the conducted work in good detail. I particularly like the design of Figure 2 to visualize the model-data aspect. The study design is relevant, as C-N models are best informed by field experiments that include measurement estimates of both C and N processes. It is also an advance over other model studies such as Meyerholt & Zaehle (2015), where only one site was used for a model-data comparison of ecosystem N recovery. However, I am not convinced that this study is a good fit for BG,

as it suffers from model limitations that make the results not meaningful to a general biogeoscientific audience, whereas the study has interesting insights for the land modelling community that are laid out in the discussion (e.g. section 4.1). Therefore, I see the paper as a fit for e.g. GMD, but cannot recommend publication in BG.

The main issue I see is that neither the standard or the adjusted version of CLM5 used here appear capable of simulating plausible C and N cycle representations, the conclusion being that they cannot presently be used to give e.g. meaningful estimates of C sink responses to N change. The authors are aware of this for the standard version and fully describe the changes they made to come up with the adjusted version that is supposed to be a better fit for the site selection. To my understanding, however, they fix a hole (unrealistic equilibrium C and N stocks) by creating another one (e.g. eliminating "denitrification"). This fixes some site specific measures, but it also creates a C-N model without a plausible N cycle. Apparently on average, live wood C:N ratios are at the level of foliage (Table 2)? The adjusted ecosystem N residence time appears rather arbitrary with a huge range (p9l18). Also, although the presentation is commendably thorough, the model formulations of key N processes are not clearly given. Since this is central to what we can expect the model to do as far as N, they deserve explicit description beyond reference to other studies (in particular, Lawrence et al. 2018 is not listed in the references). At this state it is not clear how N fixation is calculated - the model uses FUN, but Shi et al. describe that FUN is only used to determine the partitioning between uptake and symb BNF, whereas total N input uses CLM4 standard? So is BNF still NPP-based? Similarly, it is not clear how loss fluxes are determined in this study - but ecosystem N inputs and outputs a central to how N recovery is calculated. In my opinion, these model-related problems push the study more towards a well-presented model-tuning exercise - which is not bad at all and definitely needed for model evaluation, but not relevant to the broader BG readership. To this end, I disagree with the authors on some of the early discussion points: "Our study provides insight into which model assumptions are consistent, or inconsistent, with experimental results." (p13l13) - So if fixed ecosystem N inputs, unrealistic C:N

ratios and the elimination of gaseous N losses lead to a number that is consistent with experiments, does that make the assumptions correct?

Other things:

- I think the title should state that the study is about N recovery rather than C sinks. Also, I understand "decadal" to mean decade-by-decade, rather than "some experiments last over 10 years".

- Abstract p2l11 ff: It appears that for longer timescales, model plants did not acquire more than twice the experimental N recovered (23% vs 13%).

- very minor point, I was a bit confused by the order of in-text citations for multiple references in the same bracket. Since the order is not by year or by name, it is by relevance? But this can't be the case for the citations in p3l12f.?

- p6l25: There is a double "g N" in the middle.

Reference: Meyerholt, J. & Zaehle, S. The role of stoichiometric flexibility in modelling forest ecosystem responses to nitrogen fertilization. New Phytol. 208, 1042-1055, doi:10.1111/nph.13547 (2015).

---

## Referee Comment (RC2) · Stocker (Referee) · 24 Jan 2019

First, please excuse my long response time.

This paper tested a terrestrial ecosystem model with coupled C and N cycles against 13 long-term 15N tracer addition experiments at 8 temperate forest sites. It thus targets a central quantity that has received great attention and has served to synthesize the complex nature of C-N cycle interactions: Where and how much N is retained in the ecosystem? How much C is additionally stored in plants and soil per added N? Model evaluation in this respect is thus a highly welcome exercise. The model in its original version is a widely used global vegetation model and is used as a compoenent

within a coupled Earth System Model. A careful examination is thus important also for understanding the power and limits of respective ESM simulations (which will be part of CMIP6, I guess). Because this model (CLM5) overestimates N inputs and losses (too open N cycling), a re-configured version is used here, where N cycle openness is drastically reduced by reduced prescribed N deposition, artificially supressed N fixation, and artificially supressed denitrification.

Tracer studies suggest important contribution of recovery by immobilisation. Both model configurations overestimated N recovery in plants. The adjusted model version simulated N recovery in soils ok, while the original version underestimated soil N recovery. Regarding implications for C: the model has too low C:N ratios in wood and thereby underestimates the amount of C sequestered per N added, thus balancing the overestimation in simulated plant N recovery. The relevance of the two quantities (% recovery in different pools, and dC/dN) is high and the model evaluation has promise to yield useful insights into a key mechanism.

Confronting models with data is in general a very important way forward to addressing known unresolved deficiencies in models. The present paper takes a step in that direction. Therefore, I am looking forward to some (major) revisions of the present manuscript and consider that the paper (after revision) may be a valuable contribution to this field and suitable for publication in Biogeosciences. However, I have a few concerns regarding the comparability of observed and modelled quantities which need careful additional explanation and justification. In addition, I consider that the discussion of the insights gained and the recommendations given for further research are too generic and am not convinced that proposed steps will advance the field, or even the performance of the model used here. Looking at the fate of N additions also provides insights into the complex interactions between microbes and different soil compartments (surface, depth, mineral N). However, the complex behaviour of the system also points to difficulties related to the model setup for mimicking the experiments. These limitations should be addressed better and possibly alternatives explored before the

paper is re-considered for publication. Having said that, I appreciated the well-written text and clear presentation.

MAJOR

* I'm asking myself about the comparability of simulated N recovery (Eq. 1) and 15N-based results and what underlying assumptions have to be met. One is perfect mixing in the respective pool. It could be that the N added in the field enters the system in such a way as to favour accumulation in the litter by being taken up by microbes (immobilisation), and is never fully mixed within the entire mineral N pool available to the plant. In contrast, in the model it is added directly to the mineral N pool and is well mixed (by design). Once N is immobilised (in nature), this doesn't preclude that additional N (that would otherwise have fuelled immobilisation) is now available to the plant and leads to an increase in overall plant N uptake. However, this N then doesn't carry the signature. If this is indeed a possibility, then one would have to either compare total N stocks in observations analously to Eq. 1. This problem of the field-model comparison is discussed on l.16, p. 15 and it is pointed out that the tracer is added (in the field) to litter layer where immobilisation occurs, whereas in the model it's added to the mineral N pool. The suggested solution to this conceptual mismatch is making the model more complex and I'm not convinced this is a path worth taking. Wouldn't it just do the trick to add the N (in the model) to the litter pool and thus decrease its C:N ratio?

* My other concern/question is that 15N data is compared to simulated N pool size changes. The most direct comparison would be to track a 15N tracer also in the model, but I understand that this is beyond the scope here. But can the two be compared in an experimental setup, where no additional N is added as fertiliser (see entries with 0 g N yr-1 m-2 in Table 1). How is this handled in the model? The denominator in Eq. 1 is zero in this case. In that sense, I'm asking myself about what is the advantage of using tracer studies as opposed to just looking at biomass changes in response to fertilisation. Adding a clearer motivation for the approach chosen here in the intro

and/or discussion may address this question.

\* Data availability: In the introduction (p. 4, l.26), it is stated that "we first compiled a summary of 15N recovery data from long- term 15N tracer experiments that can be used to evaluate N cycling in other land models. " However, the data is not made accessible (only upon request). This aspect seriously hinders progress and I was disappointed to see it here.

\* I was a bit disappointed by the discussion where it is suggested that the inclusion of more processes (p.14, l.26) would lead to a better model-data agreement. This is a common "reflex" seen in many publications but often doesn't resolve model deficiencies that are implied by its "core", and it doesn't take into consideration possible shortcomings of the nature of model-data comparison itself. Increasing model complexity often leads to more problems and yet more calls for adding processes. Here, this suggestion is rather disconnected from the findings presented here, and I was disappointed to see it here.

\* I am intrigued by the issues of N cycle openness in the CLM5 model and would be interested to learn more about how the results for N fate presented here yield new insights into this problem and how it can be resolved. The model was re-configured here with much reduced N inputs (deposition and fixation) and an unrealistic suppression of denitrification. While this allowed authors to achieve plausible results for N recovery, it gives the impression that a wrong model behaviour was "fixed" by an unrealistic assumption. Does this study yield insights into how this could be resolved better?

\* p.9, l.17: The calculation of N residence time: Is this calculated from the transient response or after model spinup? It has to be in steady state in order to calculate residence time = stock / loss flux. The enormous difference in residence time may just be the result of reduced N losses in a transient response.

\* Sec. 3.1, p.): Quantification of how often N cycles through the system before it's lost, as the ratio of whole-ecosystem N residence time to plant N residence time. I am

suspicious of this calculation. Once it's in the soil (and stays there for a long time), it's not actually cycling. As a more insightful charecterization of the N cycle in the two alternative model configurations, I would recommend to calculate an N cycle openness similar to Cleveland et al. 2013 PNAS as the fraction of new N to total N in new production, or Nin / ( rN:C * NPP - Nresorb ), which is in steady state equal to Nout / ( rN:C * NPP - Nresorb).

* I did not learn much from the analysis by PFT. Is there a hypothesis for why PFTs would differ in their response? This part needs a motivation, otherwise it appears here a "just because we can". It also remains unclear why CLM results differ between these PFTs. What mechanisms are responsible for differences?

* Model and observational results are reported as a mean (?) and a relative standard deviation (?). The SD is calculated from year-to-year variability, but is then treated as a standard error and a p-value is calculated to test for the significance of the difference between observed and modelled values. In my understanding, this is not permissible. The model is not stochastic, thus the variability in model results is not uncertainty. I would recommend to present numbers for the mean relative bias as mean((mod-obs)/mean(obs)).

OTHER POINTS

* I liked the intuitive characterisation of the mode used here: Maybe this could be complemented to provide additional (intuitive) model understanding on the following points: How does foliage C:N and Vcmax (which is predicted separately, I guess) interact? What drives what? How is allocation (shoot:root ratio) simulated? How are N losses simulated (scaling with a mineral N pool, or with the mineralisation flux, or...)? What happens to C paid as C cost for N uptake (respired as $CO_2$)? Can the litter-soil transfer be N limited (slowing decomposition rates)? A word more on stoichiometry optimisation applied here? Is there a feedback between plant belowground C investments and SOM decomposition rates?

* In the introduction, it may be worth citing Magnani et al., 2007 and the debate on realistic dC/dN values that ensued in response.

* Fig. 2: Unclear what bars represent. Are these the observations? If not, I highly recommend adding observations here as well. Missing info in caption. What is meant by "scenario"?

* It is argued that results for only Harvard forest are shown because it provides the longest record length. I would find it highly revealing if Fig. 2 could include results for other sites too.

SPECIFIC

* p.3, l.6: Add exchanged between the land surface and the atmosphere under future scenarios of increasing CO2 and climate change.

* p.3, l.31: A soil C:N of 5 seems very low. Where did you get this number from?

* p.6, l.7: GSWP3 data: Didn't find any data under the URL given.

* p.6, l.12: Using locally measured meteorology, elevation, soil parameters, at the experimental sites? This could be quite important.

* p.7, l.16: Equally spread across days?

* p.7, l.18: Site names appear here before being introduced. Would be worth having a short paragraph at the beginning of Methods about the locations and characteristics of sites.

* p.9, l.16: Change "equivalent" to "similar in magnitude".

* p.9, l.17: Add *N* residence time.

* p.11, l.14: Better write "per unit N tracer added" (if that's correct).

* p.11, l.18: Why not use values measured in the field? Not available? What literature?

beni stocker

---

## Referee Comment (RC3) · Anonymous Referee #3 · 30 Jan 2019

This is a well written manuscript that presents an important model-data comparison exercise to identify model failures and successes in CLM5. The author's assembled an impressive data set and use this to test CLM5's ability to capture 15N tracer recoveries and dC/dN ratios across a suite of forested ecosystems. Both reviewers have provided extensive key insights and areas for improvement for this manuscript. As such, I will keep my comments brief to not duplicate their efforts.

Major Comments

Both Reviewer 2 and the authors in the discussion (Page 15 Line 19) suggest that adding the "tracer" N to the mineral soil pool may make it too accessible to plants. Why

not perform this experiment at one of the sites at least? This would allow you to identify whether the mode of N delivery is driving the high recovery seen in plants.

Both Reviewer's highlighted that turning denitrification off may make the adjusted model simulation unrealistic. I am not an expert in gaseous losses from forested ecosystems but from my understanding I believe they dwarf those of leaching losses particularly for aerobic well drained soils that are N limited. Maybe an empirical reference to support low rates of denitrification would help here. Similarly, most readers are not actively aware of reasonable values in N residence times. Can you put the differences between the default and adjusted CLM into context with literature values?

On aside here: The use of default and adjusted combined with direct and indirect make the results difficult to digest. Why not call them open and closed N cycles or observed C:N vs. modeled C:N or some similar variant that makes sense.

What is driving the variability in wood C:N in both CLM5 versions? Is this due to LUNA? How robust is the overall observed dataset in C:N ratios. Is there only a few observations per site or are there more?

The discussion highlights reasons why the default and adjusted CLM5 can capture 15N recovery rates and dC/dN ratios but fails to address whether CLM5 needs to remain as is or have a more closed N cycle. This is almost a bigger issue than where the tracer is going given that the model is somewhat able to get the right answer for the wrong reason there.

I disagree slightly with Reviewer 2 in regards to the idea that identifying model failures that could be addressed by adding in new processes is not fruitful. Many of the efforts the authors have raised to increase plant-microbial competition by adding in explicit microbial representations and interactions between plants and microbes have proven feasible in models at the ecosystem and global scale.

---

## Author Comment (AC1) · 27 Feb 2019

To the reviewers: Thank you for taking the time to provide your thoughtful feedback on our manuscript. We appreciate your support of the peer-review publication process and open-access science. Below are our responses to your comments, with the original reviewer comment in black and our response in blue—including changes we will make to our manuscript.

Reviewer 1 Comments

1. Cheng et al. present a model-data comparison of ecosystem N recovery at several temperate forest sites, using two versions of the CLM5 -land model and a number of 15N tracer field experiments. They also use the results to give estimates of ecosystem C storage responses to changes in N availability. The paper is very well written and thoroughly describes the conducted work in good detail. I particularly like the design of Figure 2 to visualize the model-data aspect. The study design is relevant, as C-N models are best informed by field experiments that include measurement estimates of both C and N processes. It is also an advance over other model studies such as Meyerholt & Zaehle (2015), where only one site was used for a model-data comparison of ecosystem N recovery. However, I am not convinced that this study is a good fit for BG, as it suffers from model limitations that make the results not meaningful to a general biogeoscientific audience, whereas the study has interesting insights for the land modelling community that are laid out in the discussion (e.g. section 4.1). Therefore, I see the paper as a fit for e.g. GMD, but cannot recommend publication in BG.

We appreciate the reviewer's recognition of our manuscript's contribution to the modeling community. Although the model we evaluated is limited in the ways it depicts C and N cycling, we believe our results are still useful to both modelers and empiricists. Below we identify some ways we, and the other reviewers, identify our manuscript as relevant to the broader biogeochemistry community.

- First, our study is the first to synthesize [15]N tracer recovery data from across multiple measurement sites over a time period of up to almost 20 years. This synthesis is useful both for empiricists and modelers interested in using field data to evaluate N cycle processes in both field and modeled ecosystems.
- Second, our study is the first to evaluate the movement of N additions, its long-term fates, and the response of temperate forest C stocks to added N in a land model (CLM5) that is part of a state-of-the-art climate model (Community Earth System Model) on a timescale longer than a few years. This model is widely used by biogeochemists, climate scientists, and policy makers—making it a particularly important model to evaluate for its carbon cycle sensitivity to N. CLM5 is also one of the few U.S. terrestrial models involved in the global Coupled Model Intercomparison Project (CMIP) and until recently, was the only one part of this project with a N cycle. Modeling experiments that more closely simulate field experiments, such as the one we present in our paper, have also been recently discussed as important additions to understanding the biogeochemistry of ecosystems, including in Wieder et al. (submitted) and Prentice et al. (2015).
- Third, *Biogeosciences*' scope states that "Experimental, conceptual, and modelling approaches are welcome." Our study directly brings together experimentalists and modelers—as well as data from both field experiments and modeling experiments—to demonstrate how measurements and

models can be used in concert to identify limitations in an Earth system model used to project future climate and biogeochemistry.

To emphasize these points, we will modify the last paragraph of the introduction in the following ways:

- Include a sentence that says "To date, there have only been a handful of individual site-level field studies that have examined the long-term fates of N additions. This study fills this gap by compiling a summary of $^{15}$N recovery data from long-term $^{15}$N tracer experiments."
- Replace the last two sentences of the introduction with the following language that better describes our contributions to both the field and modeling community: "Through this novel data-model comparison project, we provide a synthesis of long-term, ecosystem $^{15}$N addition experiments—and identify how differences in temporal dynamics of N cycling between field measurements and CLM5 lead to divergences in measured and modeled N fate and ecosystem C responses to N additions."

2. The main issue I see is that neither the standard or the adjusted version of CLM5 used here appear capable of simulating plausible C and N cycle representations, the conclusion being that they cannot presently be used to give e.g. meaningful estimates of C sink responses to N change. The authors are aware of this for the standard version and fully describe the changes they made to come up with the adjusted version that is supposed to be a better fit for the site selection. To my understanding, however, they fix a hole (unrealistic equilibrium C and N stocks) by creating another one (e.g. eliminating "denitrification"). This fixes some site specific measures, but it also creates a C-N model without a plausible N cycle.

We share the reviewer's interest in the process of denitrification, and how to improve its field quantification and model representation; both characterizations remain highly challenging and active areas of research. This flux is extremely difficult to constrain with field measurements due to both its high degree of spatiotemporal heterogeneity and the challenge of measuring $N_2$ fluxes (Groffman et al., 2006; Butterbach-Bahl et al., 2013). There is a growing body of $N_2O$ measurements, but these fluxes from temperate forests are usually extremely small (e.g., summarized by Butterbach-Bahl et al. (2013) and Liu and Greaver (2010); as averaging $0.0087+0.0025$ kg $N_2O$–N ha$^{-1}$ yr$^{-1}$ per 1 kg N ha$^{-1}$ yr$^{-1}$). These values fail to capture the larger losses of N as $N_2$, for which there are few stand-scale annual estimates. New techniques for measuring $N_2$ flux concurrent with continuous soil environmental data have allowed estimates of whole catchment denitrification for a just a few sites, yielding estimates of 0.16 - 0.26 g N m$^{-2}$ yr$^{-1}$ (Duncan et al., 2013) and ~0.39 g N m$^{-2}$ yr$^{-1}$ (Morse et al., 2015) at deciduous hardwood sites receiving 0.8-1.0 g N m$^{-2}$ yr$^{-1}$ in N deposition.

We turned the process of denitrification off in one of our model configurations for two reasons. First, these handful of field-based measurements of denitrification in temperate forests (mentioned above) reported that rates of denitrification were relatively small. Second, ecosystem N stocks are well-measured compared to gaseous N input or output fluxes, and the constraint of the observed soil N stocks could be reached only by greatly reducing or even halting denitrification coincident with reducing N fixation to better match site-level conditions (which lack symbiotic N fixers and near negligible heterotrophic N fixation; e.g., Tjepkema (1979), Roskoski (1980), Grant and Binkley (1987), Hendrickson (1990), Barkmann and Schwintzer (1998)). Similar soil N stocks could have been achieved with small rates of

both types of N gas flux, but addressing the model's overestimates of N fixation and denitrification are topics of active research and development. In the current version of the model, denitrification accounts for 99% of modeled N losses (Figure 1), which is inconsistent with relatively well-measured hydrologic N losses and $^{15}$N tracer results such as those summarized here: Catchment data from hundreds of sites across the Northeast U.S. show that $NO_3^-$ leaching amounts to <10% and up to ~50% of N deposition of 6-11 kg N ha$^{-1}$ yr$^{-1}$; loss of DON amounts to another 1 or more kg N ha$^{-1}$ yr$^{-1}$ (Aber et al., 2003). In addition, past studies that we have cited report that previous versions of CLM grossly overestimate (> 90%) denitrification both spatially and temporally (Houlton et al., 2015; Nevison et al., 2016). Rather than include this process in our adjusted model and its implausibly high estimates for these temperate forests, we turned off denitrification to test model performance with inputs and loss fluxes closer to reported measurements for these types of forests (e.g., Yanai et al. (2013) and Bernal et al. (2012)). Our two model configurations serve as sensitivity tests—so that we can understand the fate of N additions in a system with a relatively closed N cycle, and another with a highly open N cycle.

We do acknowledge both versions of the model imperfectly represent N cycle processes. However, given the popular and continued use of CLM5, CESM, their model outputs, and their use in international climate simulation comparisons, identifying the limitations to these models is important for developing appropriate interpretations and uses of these models and their projections in the biogeosciences community. Comparing the default version of CLM5 against a version with denitrification eliminated allowed us to specifically identify if, and how, excessively high N inputs and losses contributed to the drastic underestimation of N recovery in simulated forests. Considering this comment and other reviewer suggestions, we will reframe our two models as representative of two alternative scenarios—one with a highly open N cycle (model with high N inputs with denitrification on) and one with a relatively closed N cycle (more realistic N inputs with denitrification off). In addition to shifting language of "default" and "modified" models to "open" and "closed" models, we will add the following paragraph in the Methods (Section 2.3.1) to clarify why the adjusted (now "closed") version of the model was included. These sentences will read:

 "The CLM greatly underestimates measured rates of N losses to leaching and runoff in temperate forests (e.g., MacDonald et al. (2002), Aber et al. (2003), Nevison et al. (2016), Thomas et al. (2013a)). Because the CLM both underestimates leaching and overestimates N inputs and denitrification losses, we included an alternative "closed" version of the model in which N inputs were reduced to better match observations, and N losses to denitrification were correspondingly halted to allow the model to build realistic soil N pools, along with some N leaching losses (described in Sections 2.3.1. and 3.1). A model that simulates small fluxes of both heterotrophic N fixation inputs and denitrification losses might best match observations of these processes in these temperate forests (Tjepkema, 1979; Roskoski, 1980; Hendrickson, 1990; Barkmann and Schwintzer, 1998; Bernal et al., 2012; Duncan et al., 2013; Morse et al., 2015), but requires substantial model development to achieve (Thomas et al. 2013a, Houlton et al. 2015). Our alternative model included in this study is thus an oversimplification included to examine model C-N responses in ecosystems with a much more closed N cycle than the widely used version, details of which can be found in Lawrence et al. (2018). Our results below highlight that CLM is sensitive to the openness of the N cycle, an emergent property that should be the focus of future model development."

Similarly, our dC/dN calculations provide an estimate of what the model simulates to be the C response of ecosystems to N addition and how sensitive CLM5 is to changes in N cycle openness. To make this clear to the reader, we will add additional text within our results and discussion section saying that our dC/dN estimate provides a sensitivity test for how dC/dN changes with the openness of a system's N cycle through our comparison of open and closed model depictions of N cycling against our best estimate of dC/dN from field measurements. The first sentence of the section will read, "To scale and compare the effect of plant and soil N recoveries on forest C sinks between the closed configuration of CLM5 and field measurements, we estimated changes in plant, soil, and total C stocks (i.e., sum of plant and soil stocks) in response to N tracer or fertilizer additions—referred to as $(dC/dN)_{plant}, (dC/dN)_{soil}, (dC/dN)_{total}$, respectively (Fig. 4)." We will also end this section with the following caveat: "Because of existing model limitations in N cycle representation, model-estimated values of dC/dN are intended to provide a sensitivity test of how the modeling of N fates can affect model estimates of ecosystem C response to N additions relative to what is expected from field measurements."

3.  Apparently on average, live wood C:N ratios are at the level of foliage (Table 2)?

Yes, the model prognostically predicts that live wood C:N ratios are similar to leaf C:N as indicated in Table 2. We also find these C:N ratios for live wood to be too low, and discuss the importance of assembling field measurements to constrain modeled C:N ratios of plant pools in Section 4.2.

4.  The adjusted ecosystem N residence time appears rather arbitrary with a huge range (p9l18).

It is interesting to note the large range in ecosystem N residence times (note: in response to Comment 17, we will be changing this terminology to "turnover time"). However, the mean ecosystem turnover time for the default model (880 years) is similar to the field-based estimate of turnover time at Arnot Forest (Goodale, 2017) or Hubbard Brook (Yanai et al., 2013) (i.e., [7-8 tons of N ha$^{-1}$ in soils and plants] / 8-9 kg ha$^{-1}$ of N deposition = ~800-1000 years), for example. The wide range in model-estimated turnover time across sites results from ecosystem-specific differences, including variations in factors governing organic N storage in soils (e.g., texture, past disturbance). We will add this sentence into the revised manuscript: "The wide range in ecosystem N turnover time results from site-specific differences, including factors governing N storage in soils (e.g., texture, past disturbance)"

5.  Also, although the presentation is commendably thorough, the model formulations of key N processes are not clearly given. Since this is central to what we can expect the model to do as far as N, they deserve explicit description beyond reference to other studies (in particular, Lawrence et al. 2018 is not listed in the references). At this state it is not clear how N fixation is calculated - the model uses FUN, but Shi et al. describe that FUN is only used to determine the partioning between uptake and symb BNF, whereas total N input uses CLM4 standard? So is BNF still NPP-based? Similarly, it is not clear how loss fluxes are determined in this study - but ecosystem N inputs and outputs a central to how N recovery is calculated.

We will include the following additional text about how N fixation is calculated in CLM5: "Free-living biological N fixation is calculated as a function of annual evapotranspiration and added to the soil mineral N pool. Symbiotic N fixation is passed directly to the plant and depends on plant N demand, the cost of N

fixation for the plant, and soil temperature (Lawrence et al. (in review); CLM5 technical note available at https://escomp.github.io/ctsm-docs/doc/build/html/tech_note/FUN/CLM50_Tech_Note_FUN.html).”

6.  In my opinion, these model-related problems push the study more towards a well-presented model-tuning exercise - which is not bad at all and definitely needed for model evaluation, but not relevant to the broader BG readership. To this end, I disagree with the authors on some of the early discussion points: "Our study provides insight into which model assumptions are consistent, or inconsistent, with experimental results." (p13l13) - So if fixed ecosystem N inputs, unrealistic C:N ratios and the elimination of gaseous N losses lead to a number that is consistent with experiments, does that make the assumptions correct?

In our earlier response to Comment 2 (and in responses to Comments 16 and 39), switching our descriptions of the models from “default” and “adjusted” to “open” and closed” should shift emphasis to comparing two versions of the model that represent ecosystem extremes in N cycling openness, of which the closed model better matches known N input and leaching loss fluxes in these forests. These two model versions serve as a sensitivity test of how forest biogeochemistry changes when the system is more closed versus more open. We believe that pointing out the difference in N recovery between these two versions of CLM5 is of interest to a broad biogeochemistry readership to identify the contrast between observations and model simulations (Houlton et al., 2015), and because this kind of comparison encourages field data on key ecosystem processes and traits to be synthesized so that further work can be done to improve model representations of ecology.

7.  Other things: - I think the title should state that the study is about N recovery rather than C sinks. Also, I understand "decadal" to mean decade-by-decade, rather than "some experiments last over 10 years".

We changed our title to shift focus to N recovery (fate) as suggested by the reviewer, while retaining our intended scope of analysis examining C cycle responses at decadal timescales (using the Oxford English Dictionary definition of “belonging to a decade or period of ten years.”) We do think it is important to indicate that our analysis includes comparisons of N recovery at timescales relevant to meaningful C storage and to capture more than transient short-term effects. We have modified our title to read, “Decadal fates and impacts of nitrogen additions on temperate forest carbon storage: A data-model comparison.”

8.  - Abstract p2l11 ff: It appears that for longer timescales, model plants did not acquire more than twice the experimental N recovered (23% vs 13%).

We will adjust the wording to read, “In particular, the model with the closed N cycle simulated that plants acquired more than twice the amount of added N recovered in $^{15}$N tracer studies, on short timescales (CLM5: 46% ± 12%; observations: 18% ± 12%; mean across sites ± 1 standard deviation) and almost twice as much on longer timescales (CLM5: 23% ± 6%; observations: 13% ± 5%).

9.  - very minor point, I was a bit confused by the order of in-text citations for multiple references in the same bracket. Since the order is not by year or by name, it is by relevance? But this can't be the case for the citations in p3l12f.?

We will adjust the order of citations in our manuscript to be listed by year and then alphabetically by author last name.

10. - p6l25: There is a double "g N" in the middle.

We will correct this typo.

**Reviewer 2 Comments**

11. This paper tested a terrestrial ecosystem model with coupled C and N cycles against 13 long-term 15N tracer addition experiments at 8 temperate forest sites. It thus targets a central quantity that has received great attention and has served to synthesize the complex nature of C-N cycle interactions: Where and how much N is retained in the ecosystem? How much C is additionally stored in plants and soil per added N? Model evaluation in this respect is thus a highly welcome exercise. The model in its original version is a widely used global vegetation model and is used as a compoenent within a coupled Earth System Model. A careful examination is thus important also for understanding the power and limits of respective ESM simulations (which will be part of CMIP6, I guess). Because this model (CLM5) overestimates N inputs and losses (too open N cycling), a re-configured version is used here, where N cycle openness is drastically reduced by reduced prescribed N deposition, artificially supressed N fixation, and artificially supressed denitrification. Tracer studies suggest important contribution of recovery by immobilisation. Both model configurations overestimated N recovery in plants. The adjusted model version simulated N recovery in soils ok, while the original version underestimated soil N recovery. Regarding implications for C: the model has too low C:N ratios in wood and thereby underestimates the amount of C sequestered per N added, thus balancing the overestimation in simulated plant N recovery. The relevance of the two quantities (% recovery in different pools, and dC/dN) is high and the model evaluation has promise to yield useful insights into a key mechanism. Confronting models with data is in general a very important way forward to addressing known unresolved deficiencies in models. The present paper takes a step in that direction. Therefore, I am looking forward to some (major) revisions of the present manuscript and consider that the paper (after revision) may be a valuable contribution to this field and suitable for publication in Biogeosciences. However, I have a few concerns regarding the comparability of observed and modelled quantities which need careful additional explanation and justification. In addition, I consider that the discussion of the insights gained and the recommendations given for further research are too generic and am not convinced that proposed steps will advance the field, or even the performance of the model used here. Looking at the fate of N additions also provides insights into the complex interactions between microbes and different soil compartments (surface, depth, mineral N). However, the complex behaviour of the system also points to difficulties related to the model setup for mimicking the experiments. These limitations should be addressed better and possibly alternatives explored before the paper is re-considered for publication. Having said that, I appreciated the well-written text and clear presentation.

We thank the reviewer for describing how our paper adds value to the biogeochemistry community, particularly in how we identify model deficiencies in CLM5's estimates of C:N ratios and soil recovery of ecosystem N additions. To minimize reader confusion that our adjusted model is supposed to be an alternative version of CLM5, we have changed the descriptions of our two model versions from "default" and "adjusted" to "closed" and "open." In this way, our two models represent extremes in how much N recovery could occur in open and closed ecosystems (i.e., our simulations represent the sensitivity of the response to open and closed systems). We discuss the reviewer's concerns about a) the comparability of field data to model output and b) recommendations for further research and improved model performance in more detail in the reviewer's more specific comments below (Comments 12 and 13).

12. I'm asking myself about the comparability of simulated N recovery (Eq. 1) and 15Nbased results and what underlying assumptions have to be met. One is perfect mixing in the respective pool. It could be that the N added in the field enters the system in such a way as to favour accumulation in the litter by being taken up by microbes (immobilisation), and is never fully mixed within the entire mineral N pool available to the plant. In contrast, in the model it is added directly to the mineral N pool and is well mixed (by design). Once N is immobilised (in nature), this doesn't preclude that additional N (that would otherwise have fuelled immobilisation) is now available to the plant and leads to an increase in overall plant N uptake. However, this N then doesn't carry the signature. If this is indeed a possibility, then one would have to either compare total N stocks in observations analously to Eq. 1. This problem of the field-model comparison is discussed on l.16, p. 15 and it is pointed out that the tracer is added (in the field) to litter layer where immobilisation occurs, whereas in the model it's added to the mineral N pool. The suggested solution to this conceptual mismatch is making the model more complex and I'm not convinced this is a path worth taking. Wouldn't it just do the trick to add the N (in the model) to the litter pool and thus decrease its C:N ratio?

To clarify, when N deposition is vertically distributed into the soil column, it is not distributed *evenly* through the soil column. N deposition is distributed according to an *exponential* function to represent rooting depth distributions. Approximately 40% of added N deposition (including our simulated tracer and fertilizer) is distributed through the top 2 cm, and approximately 95% is distributed in the top 20 cm.

To make this clear to the reader, we will add this sentence in the methods section explaining how N deposition is distributed: "When N deposition is added to the inorganic soil N pool, it is distributed vertically through the soil column according to an exponential profile; approximately 40% of N deposition is added to the top 2 cm and approximately 95% is added to the top 20 cm." We will also modify the following sentence in the discussion section to read: "Furthermore, N additions (e.g., N deposition) are directly added to the dissolved inorganic N pool, which the model immediately distributes throughout the soil column according to an exponential profile." We will also delete this paragraph in the discussion, "In addition, N additions (e.g., N deposition) are directly added to the dissolved inorganic N pool, which the model immediately distributes throughout the soil column. However, in $^{15}$N tracer experiments, the tracer is applied directly on top of leaf litter to mimic N deposition and a vertical gradient of N use in soils forms because microbial activity and demand for N is greatest at the soil surface (Iversen et al., 2011; Li and Fahey, 2013) where fresh C inputs are greatest, C:N ratios are high, and microbes have the opportunity to rapidly capture this tracer. To capture this vertical gradient of

immobilization found in field experiments, the model could add inorganic N to the surface soil layers, only, and allow it to mix more deeply through advective or diffusive fluxes." We will now indicate the difference in how the tracer is applied in the model and in the field with the following sentence located two paragraphs above: "Although the model and field experiments differ in how they apply the N tracer to soil (directly into the inorganic soil N pool versus on top of the litter, respectively), the large magnitude of the observed soil N sink and the model's poor ability to reproduce it suggests that modeling a stronger soil immobilization sink should be a priority."

With the current model structure, we cannot add the tracer or fertilizer to the model's litter pool because it does not contain an inorganic N pool. In addition, if we added the tracer or fertilizer only to the top 2 cm of the soil, this change could have consequences on how N moves through the soil over the course of years—affecting the movement of mineralized N and leaching. Currently, the model has a diffusivity coefficient of 1 $cm^2$ $yr^{-1}$, which means it would take approximately 100 years for N to move down 10 cm. The exponential profile in the model helps to counter the model's minimal representation of vertical movement of N through the soil. While adding an explicit inorganic N pool to the litter, or explicitly capturing the vertical movement of N through soils, could be important, including them in the model with rigorous evaluation is beyond the scope of this study.

13. My other concern/question is that 15N data is compared to simulated N pool size changes. The most direct comparison would be to track a 15N tracer also in the model, but I understand that this is beyond the scope here. But can the two be compared in an experimental setup, where no additional N is added as fertiliser (see entries with 0 g N yr-1 m-2 in Table 1). How is this handled in the model? The denominator in Eq. 1 is zero in this case. In that sense, I'm asking myself about what is the advantage of using tracer studies as opposed to just looking at biomass changes in response to fertilisation. Adding a clearer motivation for the approach chosen here in the intro and/or discussion may address this question.

In Table 1, the "amount of N fertilizer" refers only to the added amount of fertilizer applied in the field— which we used in our simulations of fertilized sites. To simulate sites with no additional fertilizer, we used the approach in Thomas et al. (2013b) where a small amount of N (0.5 g N $m^{-2}$ $yr^{-1}$) was added into the soil inorganic N pool, as described in our Methods section. Because Thomas et al. (2013b) used an older version of CLM, we also conducted a sensitivity analysis (Figure S1-S2) to confirm this amount as the smallest amount of N we could apply in our simulations while recovering a reasonable amount of that added N (which we refer to as the model tracer). In this way, our denominator in Equation 1 is not 0 g $m^{-2}$ $year^{-1}$, but 0.5 g N $m^{-2}$ $yr^{-1}$ (or 0.5 g N $m^{-2}$ $yr^{-1}$ + differences in N fixation between the tracer and control simulation in the open version of the model).

We focused on simulating field-based tracer studies in order to leverage [15]N datasets to evaluate CLM's performance in simulating not only C sinks, but also the fate of added N. We find that the model actually simulates plausible C storage, but because of compensating errors in its simulation of N retention and C:N stoichiometry—errors that would not be identified if we solely examined the C storage term. Field-based tracer studies provide an advantage over the method of measuring biomass changes in response to N deposition in situations where N deposition or fertilization rates are small, partly because the tracer fate can be measured much more precisely than changes in either C or N stocks. That is, soil C, and especially

N stocks, are large and heterogenous relative to small changes in these stocks. Tracer studies, however, can identify the accumulation of N in soils even with large background N stocks.

We will add additional language about this motivation at the beginning of the discussion section, which will read: "This study compares estimates of ecosystem recovery of N additions between CLM5—a land model with coupled C and N cycles—and long-term $^{15}$N tracer experiments in temperate deciduous and evergreen forests. Compared to other methods, $^{15}$N tracer experiments provide a precise way of tracing the movement of N both between ecosystem pools and through time. In addition, changes in soil N stocks in response to N deposition are usually small relative to the background size and heterogeneity of soil N stocks and thus, difficult to capture in field measurements over time. Tracers circumvent this issue and provide the opportunity to capture small changes in soil and plant N stocks so we can identify key ecological processes and ecosystem traits that drive variability in ecosystem N recovery."

14. Data availability: In the introduction (p. 4, l.26), it is stated that "we first compiled a summary of 15N recovery data from long- term 15N tracer experiments that can be used to evaluate N cycling in other land models. " However, the data is not made accessible (only upon request). This aspect seriously hinders progress and I was disappointed to see it here.

The data we compiled and used in this manuscript are listed in the supplement, Table S1. There were a few sites where the data are listed as "available on request" because the most recent data are currently in the process of being published. We will add these numbers into our manuscript if those papers are published before our revised manuscript is available online. Otherwise, we will update this manuscript with an addendum when those numbers are published. We will also update the sentence in the Data Availability section to read, "$^{15}$N tracer data, along with key citations, are available in Table S1."

15. I was a bit disappointed by the discussion where it is suggested that the inclusion of more processes (p.14, l.26) would lead to a better model-data agreement. This is a common "reflex" seen in many publications but often doesn't resolve model deficiencies that are implied by its "core", and it doesn't take into consideration possible shortcomings of the nature of model-data comparison itself. Increasing model complexity often leads to more problems and yet more calls for adding processes. Here, this suggestion is rather disconnected from the findings presented here, and I was disappointed to see it here.

Our suggestions sought to identify and explain a few specific processes that we considered most likely to improve model performance in simulating the plant-soil dynamics that the model failed to capture here, such as a) improving mycorrhizal associations (which in part have been added to CLM5), b) adding mechanisms for priming, and c) adding an organic N pool that plants can access. These three sets of processes have all been previously identified as potentially key to N cycling (as indicated by the citations we included in our discussion section), either through modeling or empirical experiments. In Comment 41, Reviewer 3 also discusses that our suggestions have been shown to be feasible to incorporate into ecosystem and global models. As such, we would like to keep this paragraph (and the one that follows, which details more specifically why we think these three components could address model limitations) in our discussion section. We will also add additional citations, including Tang and Riley (2014) and (Zhu et al., 2016).

However, we do recognize the important balance between adding model complexity at the expense of model certainty in large-scale terrestrial land models, particularly if an added process or parameter is not well-measured. We will include additional discussion of this tradeoff at the end of this discussion section. In particular we plan to include the following: "However, additions of parameters or process-based representation of ecological processes can add uncertainty to model projections. To limit this uncertainty, new (and existing) model representations should be designed and evaluated using robust and representative, process-based datasets—as discussed in modeling papers, including Prentice et al. (2015), Lovenduski and Bonan (2017), Lombardozzi et al. (2018), and (Sulman et al., 2018).

16. I am intrigued by the issues of N cycle openness in the CLM5 model and would be interested to learn more about how the results for N fate presented here yield new insights into this problem and how it can be resolved. The model was re-configured here with much reduced N inputs (deposition and fixation) and an unrealistic suppression of denitrification. While this allowed authors to achieve plausible results for N recovery, it gives the impression that a wrong model behaviour was "fixed" by an unrealistic assumption. Does this study yield insights into how this could be resolved better?

We share this interest, and now present our results as two extremes of N cycle openness. In response to this comment and Comment 2 above, we have provided additional explanation on the rationale for vastly reducing the default model's denitrification rates, and we recognize that the reality about these assumptions lies somewhere between the two model scenarios we present. The default model largely fails to capture the plant-soil N competition indicated by the $^{15}$N tracer studies, and this failure is why we propose a few potential steps for their improvement in our Discussion section (also discussed in response to Comment 15). The two model configurations should be viewed as a sensitivity test of the response of open and closed systems.

Our study, like prior analysis of earlier versions of CLM (Koven et al., 2013; Thomas et al., 2013a; Thomas et al., 2013b; Houlton et al., 2015) demonstrates that denitrification, N deposition, and N fixation all need to be addressed. Although our comparison of two model versions do not explicitly identify a way to fix the prognostic representation of denitrification and N fixation, our study does indicate that adjustments to losses are best done in concert with tests that adjust inputs. We emphasize the effect of N cycle openness on N fates in the discussion section, as discussed in our response to Comment 40.

17. p.9, l.17: The calculation of N residence time: Is this calculated from the transient response or after model spinup? It has to be in steady state in order to calculate residence time = stock / loss flux. The enormous difference in residence time may just be the result of reduced N losses in a transient response.

The calculations of N residence time that we report in the results section are calculated from the end of our transient simulations. However, we also include calculations of N residence time done at steady state in Table S2. After spin up, ecosystem N residence times are similar in magnitude to those calculated at the end of the transient simulations. To avoid confusion about whether our calculations are done in steady state, we will change terminology of residence time to turnover time. The variation in turnover time across sites was addressed in our response to Comment 4.

18. Sec. 3.1, p.): Quantification of how often N cycles through the system before it's lost, as the ratio of whole-ecosystem N residence time to plant N residence time. I am suspicious of this calculation. Once it's in the soil (and stays there for a long time), it's not actually cycling. As a more insightful charecterization of the N cycle in the two alternative model configurations, I would recommend to calculate an N cycle openness similar to Cleveland et al. 2013 PNAS as the fraction of new N to total N in new production, or Nin / ( rN:C * NPP - Nresorb ), which is in steady state equal to Nout / ( rN:C * NPP - Nresorb).

We thank the reviewer for suggesting this alternative metric. The reviewer is correct of course, that once N is in the soil, it can stay there a long time, but not forever (see comment above). The metric we use in our paper (i.e., the turnover time of the N in the ecosystem divided by that in the vegetation) helps meet the goals of our paper by distinguishing very clearly the differences in N cycling between the closed and opened systems (formerly adjusted and default models). Thus, we would like to continue using this metric so that we can quantitatively demonstrate how open or closed these two model configurations are relative to each other. The Cleveland et al. metric does not seem to clearly distinguish between the open and closed systems in the model because Nin is the same in the open and closed systems, and both systems have similar rates of NPP. A simple interpretation of our metric is how many times on average N cycles through plants before it is lost from the ecosystem.

19. I did not learn much from the analysis by PFT. Is there a hypothesis for why PFTs would differ in their response? This part needs a motivation, otherwise it appears here a "just because we can". It also remains unclear why CLM results differ between these PFTs. What mechanisms are responsible for differences?

We hypothesized that PFTs would differ in their N fates because of differences in growing season length and the C:N ratios of foliage, which would influence the C:N ratio of litter as well as rates of decomposition and transfer of N to litter and soil pools with longer turnover times. The hypothesis that PFT traits influence N recovery through immobilization rates is supported by the larger portion of N immediately recovered in soils compared to plants (Fig. 5b v. 5a).

We will add these two sentences to the beginning of the PFT section to motivate this analysis. The sentences will read: "In the field, forest types might respond to N deposition differently because of differences in their plant and ecosystem traits (Cornelissen 1996). In CLM5, the evergreen and deciduous PFTs differ especially in foliage C:N (see Table 2) and timing of plant N demand, which should alter decomposition and N mineralization."

20. Model and observational results are reported as a mean (?) and a relative standard deviation (?). The SD is calculated from year-to-year variability, but is then treated as a standard error and a p-value is calculated to test for the significance of the difference between observed and modelled values. In my understanding, this is not permissible. The model is not stochastic, thus the variability in model results is not uncertainty. I would recommend to present numbers for the mean relative bias as mean((modobs)/mean(obs)).

In Figure 3, the bars represent the distance between the 1st and 3rd interquartile range of the data (which include the measured or modeled recoveries across sites). Similarly, in Figure 4, error bars represent 1 standard deviation across the $^{15}$N data measured across the field sites. Although each site measured recoveries at different timepoints after the tracer was added, the standard deviation in Figure 4 predominately represents variation across sites rather than temporal variation. In these figures and with our ANOVAs, we are identifying whether differences exist between groups of data. We agree that calculating relative bias is a nice approach to describing how far away the model is from the measurements. However, part of the goal of our paper was to synthesize and illustrate the temporal patterns in N recovery across the decadal timescale for the field sites. Switching to mean relative bias would remove the presentation of the absolute values of the field data, and reduce the accessibility of the field results to readers—particularly those interested in the empirical data.

In our figure captions, we will more clearly define that the standard deviations or interquartile range represent cross-site variation and that we are comparing differences between groups, as opposed to comparing the means of two groups.

21. I liked the intuitive characterisation of the mode used here: Maybe this could be complemented to provide additional (intuitive) model understanding on the following points: How does foliage C:N and Vcmax (which is predicted separately, I guess) interact? What drives what? How is allocation (shoot:root ratio) simulated? How are N losses simulated (scaling with a mineral N pool, or with the mineralisation flux, or. . .)? What happens to C paid as C cost for N uptake (respired as CO2)? Can the litter-soil transfer be N limited (slowing decomposition rates)? A word more on stoichiometry optimisation applied here? Is there a feedback between plant belowground C investments and SOM decomposition rates?

We will add additional information about the model structure to the methods section, including the following sentences (Note: We felt that discussing the specifics of allocation would distract the reader from the main points of our study, and have added a reference to the CLM5 documentation with information about allocation. Much of our cited literature in the methods also go into more details about these model representations):
- "Denitrification only occurs in the anoxic portion of the soil and is constrained by decomposition and the availability of nitrate."
- Environmental conditions influence a leaf's photosynthetic capacity. "Specifically, the maximum rate of carboxylation ($V_{c,max}$) is influenced by the amount of leaf N allocated for carboxylation, as well as day length and season."
- "The amount of N that is allocated to individual sub-plant pools is determined based on a fixed set of allometric ratios and the amount of N the plant has for new growth. Additional details on how stoichiometry is optimized can be found in the CLM5 documentation referenced below.
- Microbial decomposition of litter is implicitly represented through the transfer of N from the litter pools to the soil pools. During decomposition, carbon is respired and rates of decomposition are limited by soil moisture, soil temperature, and N availability.

We will also clarify in our methods section that the C costs are respired by editing the sentence to read, "plants pay C costs (which are respired) for acquiring N from symbiotic N fixation…"

22. In the introduction, it may be worth citing Magnani et al., 2007 and the debate on realistic dC/dN values that ensued in response.

Magnani et al., 2007 did spur an interesting debate about what realistic dC/dN values are. However, subsequent papers, including Sutton et al. (2008) addressed most of the problems that were part of the Magnani et al. 2007 analysis that spurred so much debate.

We will modify the end of the first paragraph to introduce the importance of calculating dC/dN. The last sentence of the first paragraph will read, "Thus, evaluating model representations of N cycling is critical to increasing confidence in projections of ecosystem C responses to changes in N inputs (dC/dN; Sutton et al., 2008) and how dC/dN influences the size of the terrestrial C sink over the 21st century."

23. Fig. 2: Unclear what bars represent. Are these the observations? If not, I highly recommend adding observations here as well. Missing info in caption. What is meant by "scenario"?

The stacked bars do represent observations. We will change the figure caption description from "field measurements" to "observations." "Scenario" refers to each simulation for a site and specific N input. We will remove "For each scenario" to avoid confusion.

24. It is argued that results for only Harvard forest are shown because it provides the longest record length. I would find it highly revealing if Fig. 2 could include results for other sites too.

Our previous attempts to incorporate all sites into one figure made the plots difficult to interpret, but results for all of the sites are included in Figures S3-S5. We modified the figure caption to make this information clearer to the reader. The sentence now reads, "Plots of recoveries at all other sites and for the default version of CLM5 at Harvard Forest are in Figs. S3-S5."

25. p.3, l.6: Add exchanged between the land surface and the atmosphere under future scenarios of increasing CO2 and climate change.

We will add "under future scenarios of increasing $CO_2$ and climate change" to this sentence as suggested by the reviewer.

26. p.3, l.31: A soil C:N of 5 seems very low. Where did you get this number from?

C:N ratios of 5 have been measured in deep, low C soils at Arnot Forest (Goodale, 2017), while the 25 is lower than the value from Nadelhoffer et al. (1999) which was reported as an upper bound for the forest floor. We will add the Arnot Forest citation to this sentence.

27. p.6, l.7: GSWP3 data: Didn't find any data under the URL given.

We will revise the link to the data http://search.diasjp.net/en/dataset/GSWP3_EXP1_Forcing and add its citation to this dataset: Hyungjun Kim. (2017). *Global Soil Wetness Project Phase 3 Atmospheric Boundary Conditions (Experiment 1)* [Data set]. Data Integration and Analysis System (DIAS).

28. p.6, l.12: Using locally measured meteorology, elevation, soil parameters, at the experimental sites? This could be quite important.

For each forest, we used local meteorology prescribed from the GSWP3 dataset (see Comment 27), and elevation soil characteristics derived from high-resolution input datasets as listed in the CLM5 documentation. Because not all of our forest sites had meteorological tower data, we used the model's input variables for these grid cells in order to maintain consistency.

29. p.7, l.16: Equally spread across days?

Yes. We will add this information into the methods section. The sentence will read, "...we applied the N "tracer" equally across days during April through September to capture the most active portion of the growing season."

30. p.7, l.18: Site names appear here before being introduced. Would be worth having a short paragraph at the beginning of Methods about the locations and characteristics of sites.

The beginning of our Methods (Section 2.1) described the Table we included with locations and characteristics of these sites. To help link that section to this portion of the Methods section, we will reference the Table in section 2.2. The sentence will read: "In CLM5, we ran sensitivity tests for two of our eight model sites (see Table 1 for a full list, including site names), an old growth forest (Alptal) and a younger forest (Harvard NET), which confirmed that the smallest amount of N we could apply while maintaining realistic ecosystem N recovery responses at both sites was 0.5 g N m$^{-2}$ y$^{-1}$ (Fig. S1), consistent with Thomas et al. 2013b."

31. p.9, l.16: Change "equivalent" to "similar in magnitude".

We will make this change in the text.

32. p.9, l.17: Add *N* residence time.

We will make this change in the text.

33. p.11, l.14: Better write "per unit N tracer added" (if that's correct).

We will revise the sentence to better describe Eqn. 2. Upon editing, the sentence will read, "For the model, annual dC/dN values were computed directly (i.e., "direct approach") as the difference between the total plant or total soil C stocks between the baseline and "tracer" (or fertilizer) simulation divided by the difference in the amount of cumulatively added N between the two simulations (Eqn. 2)."

34. p.11, l.18: Why not use values measured in the field? Not available? What literature?

The literature-based estimate we mentioned in this sentence refers to published field literature from the sites we analyzed in our paper. We adjusted the sentence to better reflect that information. The mean of the published values are listed in Table 2, with individual site values listed in Table S3. The sentence now reads, "...using the measured $^{15}$N recovery in each pool and published field-measured values of C:N for that site's particular pool (Table 2, Eqn. 3)."

Reviewer 3

35. This is a well written manuscript that presents an important model-data comparison exercise to identify model failures and successes in CLM5. The author's assembled an impressive data set and use this to test CLM5's ability to capture 15N tracer recoveries and dC/dN ratios across a suite of forested ecosystems. Both reviewers have provided extensive key insights and areas for improvement for this manuscript. As such, I will keep my comments brief to not duplicate their efforts. Both Reviewer 2 and the authors in the discussion (Page 15 Line 19) suggest that adding the "tracer" N to the mineral soil pool may make it too accessible to plants. Why not perform this experiment at one of the sites at least? This would allow you to identify whether the mode of N delivery is driving the high recovery seen in plants.

We addressed in depth the difficulty of developing a model version where we could add the "tracer" only to the litter pool or top surface layers of the soil in our response to Comment 12. Overall, approximately 50% of added N deposition (including a tracer and fertilizer) is distributed into the top 2 centimeters (and approximately 95% is in the top 20 centimeters), and so most of the added tracer or fertilizer is available in the top layers of the soil. Attempts to only add the tracer into the top layer of the soil would require significant modeling changes and evaluation that are beyond the scope of this study

36. Both Reviewer's highlighted that turning denitrification off may make the adjusted model simulation unrealistic. I am not an expert in gaseous losses from forested ecosystems but from my understanding I believe they dwarf those of leaching losses particularly for aerobic well drained soils that are N limited. Maybe an empirical reference to support low rates of denitrification would help here.

We addressed questions about turning off denitrification in our response to Comment 2. Briefly, although simulated losses of N to denitrification dwarf leaching losses in various versions of CLM (Koven et al., 2013; Thomas et al., 2013a; Houlton et al., 2015; Nevison et al., 2016), this particular imbalance has been identified as a problem by these and related studies. While few field measurements exist to quantitatively constrain full denitrification losses, leaching losses are relatively well-measured and vastly exceed current model predictions (see response above for Comment #2 for more details and further citations).

37. Similarly, most readers are not actively aware of reasonable values in N residence times. Can you put the differences between the default and adjusted CLM into context with literature values?

It's true that residence times of N (note: We are changing this terminology to "turnover times" as discussed in our response to Comment 17) in ecosystems are rarely computed or reported for field studies. Nonetheless, they can be calculated (as described in our response to Comment 4) for each site using the soil N stocks and rate of major input or output fluxes (for these systems, N deposition). Because we are changing our descriptions of the two model versions from "default" and "adjusted" to "open" and

"closed," the comparisons of the turnover times will serve to show how differently N cycles through the two model versions—as opposed to representing a comparison to what we might expect to find in the field. Adding a range of values calculated from sites may detract from the main message of this section that the two models represent extremes of N cycling (open N cycle and closed N cycle).

38. On aside here: The use of default and adjusted combined with direct and indirect make the results difficult to digest. Why not call them open and closed N cycles or observed C:N vs. modeled C:N or some similar variant that makes sense.

We agree with the reviewer that calling our two models "open" and "closed" could be easier for some readers to digest. It also allows us to describe how our two models are representative of either extreme of N cycling in ecosystems and to reduce confusion about whether the "adjusted" model is supposed to be "better" with denitrification turned off. We will correspondingly change references to "default" and "adjusted" to "open" and "closed" in revisions.

39. What is driving the variability in wood C:N in both CLM5 versions? Is this due to LUNA? How robust is the overall observed dataset in C:N ratios. Is there only a few observations per site or are there more?

The FlexCN module in CLM5 prognostically calculates the C:N of each plant tissue (Ghimire et al., 2016); thus with the FUN module (Shi et al., 2015), when foliar C:N increases, the plant spends more C on acquiring N (Lawrence et al., in review). CLM5 applies a tissue C allocation to specific plant pools that is parameterized for each plant functional type (see Lawrence et al. (in review); CN Allocation section in Lawrence et al. (2018)).

The number of observations for wood C:N ratios varies by site and depends on how many tree cores were taken. For example, at Arnot Forest, 40-50+ wood samples were collected during each sampling date (Goodale, 2017). Table S3 includes citations for the tracer experiments, which include methods for sampling trees. We will modify the last sentence of the caption for Table 2 to read: "Site-reported or estimated values for C:N ratios for each site and their references (which describe the sampling methods for each pool), are in Table S3."

40. The discussion highlights reasons why the default and adjusted CLM5 can capture 15N recovery rates and dC/dN ratios but fails to address whether CLM5 needs to remain as is or have a more closed N cycle. This is almost a bigger issue than where the tracer is going given that the model is somewhat able to get the right answer for the wrong reason there.

We agree with the reviewer that one of the most important issues our data-model comparison points out is unrealistic fluxes of N inputs and denitrification. From our analysis, we provide additional support for needing to create more accurate input datasets for N deposition as well as improving prognostic calculations of N fixation and denitrification. We have modified the beginning of the discussion to add a section that highlights this. This section will read as:

**4.1 Modeling Ecosystem Inputs and Losses**

In our analysis of ecosystems with open N cycles (which represent the default configuration of CLM5), we identified that CLM5 continues to have large biases in N losses (Fig. 1)—similar to previous versions of CLM (Thomas et al., 2013b; Nevison et al., 2016; Koven et al., 2013; Houlton et al., 2015). Specifically, CLM5 has unrealistically high rates of denitrification and low rates of N leaching and runoff compared to field measurements, as reviewed in the above-mentioned analyses. We also identified that pre-industrial N deposition in CLM5 was higher than expected from reconstructions (Fakhraei et al., 2016; Holland et al., 1999) for the Northeast United States and parts of Europe (Fig. 1). Although both model configurations shared some similar responses to N additions, the ecosystems with higher N deposition and denitrification fluxes typically a) had less total ecosystem recovery of N inputs than ecosystems that had more closed N cycles and b) underestimated long-term soil N recovery compared to observations. In adjusting N inputs and losses to better match field expectations, many of the simulated ecosystem stocks and fluxes (i.e., plant N, soil N, plant C, leaf area, and ANPP) were similar to those estimated by the default configuration of the model and to observations (Table 3). Given that the openness of an ecosystem's N cycle affects its recovery of N inputs on the decadal timescale, we suggest that future model development not only test new mechanistic representations of N fixation and denitrification, but do so in concert with modified N input datasets to ensure that both inputs and losses capture field expectations.

41. I disagree slightly with Reviewer 2 in regards to the idea that identifying model failures that could be addressed by adding in new processes is not fruitful. Many of the efforts the authors have raised to increase plant-microbial competition by adding in explicit microbial representations and interactions between plants and microbes have proven feasible in models at the ecosystem and global scale.

We appreciate the reviewer's perspective about new processes in models. We discuss the balance between adding model complexity and minimizing model uncertainty in our response to Comment 15.